 eLife

# Structural basis of malaria transmission blockade by a monoclonal antibody to gamete fusogen HAP2

Juan Feng[1], Xianchi Dong[2], Adam DeCosta[1], Yang Su[1], Fiona Angrisano[3], Katarzyna A Sala[4], Andrew M Blagborough[4], Chafen Lu[5]*, Timothy A Springer[1]*

[1]Program in Cellular and Molecular Medicine, Boston Children's Hospital, Boston, United States; [2]School of Life Sciences, Nanjing University, Nanjing, China; [3]Burnet Institute, Melbourne, Australia; [4]Department of Pathology, University of Cambridge, Cambridge, United Kingdom; [5]Boston Children's Hospital, Boston, United States

**Abstract** HAP2 is a transmembrane gamete fusogen found in multiple eukaryotic kingdoms and is structurally homologous to viral class II fusogens. Studies in *Plasmodium* have suggested that HAP2 is an attractive target for vaccines that block transmission of malaria. HAP2 has three extracellular domains, arranged in the order D2, D1, and D3. Here, we report monoclonal antibodies against the D3 fragment of *Plasmodium berghei* HAP2 and crystal structures of D3 in complex with Fab fragments of two of these antibodies, one of which blocks fertilization of *Plasmodium berghei* in vitro and transmission of malaria in mosquitoes. We also show how this Fab binds the complete HAP2 ectodomain with electron microscopy. The two antibodies cross-react with HAP2 among multiple plasmodial species. Our characterization of the *Plasmodium* D3 structure, HAP2 ectodomain architecture, and mechanism of inhibition provide insights for the development of a vaccine to block malaria transmission.

**\*For correspondence:**
lu@crystal.harvard.edu (CL);
springer@crystal.harvard.edu (TAS)

**Competing interest:** The authors declare that no competing interests exist.

## Editor's evaluation

This study describes the generation of monoclonal antibodies targeting the fusogen HAP2 from the malaria parasite that is required for parasite transmission to mosquitoes. Using structural approaches in combination with biophysical methods, the authors provide insights into the differences in transmission blocking potencies for different monoclonal antibodies, which may inform the design of HAP2 as a potential vaccine candidate.

## Introduction

HAP2, a gamete fusogen required for fertilization, is found in eukaryotic plant, metazoan, and protozoan kingdoms (*Clark, 2018*). Crystal structures are known of HAP2 from the unicellular green alga, *Chlamydomonas reinhardti*; a model plant, *Arabidopsis thaliana*; and a protozoan parasite, *Trypanosoma cruzi* (*Fédry et al., 2017*; *Feng et al., 2018*; *Fedry et al., 2018*; *Baquero et al., 2019*). HAP2 can form a trimeric structure that is structurally homologous to the fusion state of viral class II fusion proteins. These fusogens have three extracellular domains, arranged in the order D2, D1, and D3, a disordered stem region, a single-pass transmembrane domain, and a cytoplasmic domain, and are expressed on the plasma membrane of male gametes or the envelope of viruses (*Figure 1*).

HAP2 is required for fertilization in *Plasmodium* parasites (*Hirai et al., 2008*; *Liu et al., 2008*) and is thus an attractive target for vaccines that block transmission of malaria. Gamete fertilization takes place in the mosquito midgut in a blood meal taken from an infected individual. Blocking fertilization

**Figure 1.** Diagram of HAP2 pre-fusion monomer, trimerization and membrane fusion. The model is based on postfusion structures of HAP2 referenced in the text, their similarity to postfusion structures of viral class II proteins, models for the conversion of prefusion to postfusion class II fusion proteins (*Kielian and Rey, 2006*), and work showing that D1 and D2 are sufficient for trimer formation (*Liao et al., 2010*), which suggests that the last step may be concerted foldback of D3 over D1 and D2 and fusion of the macrogamete and microgamete plasma membranes. (**A**) pre-fusion monomer on the surface of male gametes, (**B**) extended trimeric intermediate, and (**C**) post-fusion trimer and fusion of gamete membranes. Domains are colored. Fusion loops at the tip of D2 are shown as '*'. Antibodies to D3, shown as Y shapes, may agglutinate microgametes, trigger antibody Fc-dependent effector functions, or neutralize HAP2 function by interfering with trimer formation and gamete membrane fusion.

prevents transmission to the individual from which the mosquito takes its second, final blood meal. *Plasmodium* gametes emerge from infected erythrocytes well before digestion of the blood meal; specific antibodies present in the blood meal will react with and can agglutinate or otherwise neutralize gametes (*Angrisano et al., 2017*; *Blagborough and Sinden, 2009*; *Qiu et al., 2020*; *Graves et al., 1985*). HAP2 is present on male gametes (microgametes) and has hydrophobic fusion loops at the tip of D2 that insert into the plasma membrane of the female macrogamete and trigger trimerization and structural rearrangements of HAP2 to the fusion state, which mediates membrane fusion and merging of the cytoplasm of the two gametes through a fusion pore (*Kielian and Rey, 2006*; *Figure 1*). To mediate fusion, D3 of HAP2 folds over an inner trimeric core composed of D1 and D2 (*Figure 1*). We reasoned that antibodies to D3 of HAP2 that blocked the interface formed with D1 and D2 in the fusion state might block fusion, as has been demonstrated with viral fusogens (*Austin et al., 2012*; *Zhao et al., 2016*; *Li et al., 2018*). Thus, we hypothesized that HAP2 D3 antibody taken up in a blood meal could inhibit fertilization and infection of the mosquito and subsequent transmission to humans (*Delves et al., 2018*; *Figure 1*).

Immunization of mice with HAP2 fragments or putative HAP2 fusion loop peptides has proved the concept that antibodies to different domains of HAP2 can, to different extents, block transmission of *Plasmodium berghei* or *P. vivax*, species that infect rodents and humans, respectively (*Angrisano et al., 2017*; *Blagborough and Sinden, 2009*; *Qiu et al., 2020*). However, we are far from having an optimized HAP2 immunogen. Furthermore, the high-sequence divergence of HAP2 in *Plasmodium* species from structurally characterized HAP2 in other phyla emphasizes the importance of *Plasmodium* HAP2 structures for rational vaccine design. Here, to advance a vaccine that would block malaria transmission, we have expressed a D3 fragment of *P. berghei* HAP2, raised monoclonal antibodies, crystallized D3 in complex with Fab fragments of two antibodies, and examined Fab complexes with the complete HAP2 ectodomain (D1-D3) by negative stain electron microscopy (EM). Furthermore, we show that one of these antibodies potently blocks gamete fertilization and transmission and thus for the first time that a monoclonal antibody to HAP2 can block transmission. We define the advantages, as well as the limitations, of using the D3 domain of HAP2 as an immunogen for transmission blockade. Moreover, the two structurally characterized antibodies have cross-reactivity with HAP2s among multiple plasmodial species that can cause malaria in humans. The insights into *Plasmodium* D3 structure, HAP2 ectodomain architecture, and mechanism of inhibition are important steps toward the development of a vaccine to block malaria transmission.

## Results

### *P. berghei* HAP2 D3 elicits antibodies that cross react with human malaria pathogens

Apicomplexans such as *Plasmodium* are eukaryotes that have extracellular proteins that are disulfide-linked and glycosylated; therefore, we produced HAP2 proteins in insect and mammalian cells which are competent for such modifications. C-mannosylation and O-glycosylation in *Plasmodium* have been verified; however, while *Plasmodium* species have been suggested to have unusually short N-glycans, N-glycosylation has yet to be verified with any specific protein (*Bushkin et al., 2010*; *Macedo de et al., 2010*; *Bandini et al., 2019*; *Swearingen et al., 2016*; *Swearingen et al., 2019*). To elicit antibodies, we expressed in *Drosophila* S2 cells a *P. berghei* HAP2 D3 construct (residues 477–621, containing four putative N-glycosylation sites) containing a C-terminal His tag. After Ni-affinity chromatography, the material was nearly homogenous, as shown by SDS-PAGE and Coomassie blue staining (*Figure 2A*, lane 1). Shaving the N-glycans with endoglycosidase D (Endo D) followed by gel filtration decreased size and heterogeneity in SDS-PAGE and confirmed N-glycosylation of D3 (*Figure 2A*, lane 2).

To produce monoclonal antibodies (mAbs), mice were immunized with the glycan-shaved D3. Antibodies elicited to the shaved protein, which contains one N-glycan residue at each N-glycosylation site, would be expected to react with D3 without being influenced by its glycosylation status. Five IgG mAbs reacted with the immunogen with nanomolar EC50 values in ELISA (*Figure 2B*). Surface plasmon resonance (SPR) showed that Fabs of two mAbs, 2/6.14 and 2/1.12, bind with affinities of 5–10 nM (*Figure 2—figure supplement 1*). The antibodies also immunoprecipitated a shorter D3 (502–617) with all putative N-glycosylation sites mutated out (*Figure 2C*, top panel). Immunoprecipitation was comparable to the His-tag antibody, showing that the epitopes of the 5 mAbs resided within residues 502–617 and were unaffected by N-glycosylation. One antibody (1/5.13) that reacted with the C-terminal His tag and is comparable in specificity and sensitivity to commercial His tag antibodies is described in Materials and methods (*Figure 2—figure supplement 2*).

HAP2 D3 was also expressed and shaved with Endo D from *Plasmodium* species that can infect humans: *P. knowlesi*, *P. vivax*, *P. malariae*, *P. ovale*, and *P. falciparum* (*Figure 2A*). As their D3 domains share 60–70% amino acid sequence identity with *P. berghei*, we tested cross-reactivity with the mAbs to *P. berghei* D3 using immunoprecipitation (*Figure 2C*). We quantified immunoprecipitation (*Figure 2D*) and also measured cross-reactivity using ELISA (*Figure 2E* and *Figure 2—figure supplement 3*). mAbs 2/6.14 and 2/1.12 were especially efficient at recognizing D3 from multiple species (*Figure 2C*). mAb 2/6.14 pulled down about 70%, 30%, and 20% of *P. knowlesi*, *P. ovale*, and *P. malariae* D3, respectively, and mAb 2/1.12 was even more reactive, pulling down 60–100% of D3 from all species except *P. falciparum* D3 (*Figure 2C and D*). In contrast, mAb 2/1.40 showed strong reactivity to *P. ovale*, but not to D3 of other species, while mAbs 2/3.3 and 2/4.36 were highly specific for *P. berghei* (*Figure 2C and D*). mAbs 2/6.14 and 2/1.12 bound all species of D3 tested, as shown by ELISA EC50 values in the 1–1000 nM range (*Figure 2E* and *Figure 2—figure supplement 3*). Strong binding of *P. knowlesi* D3 to both 2/6.14 and 2/1.12 was observed with EC50 of <10 nM.

### Antibody reactivity with gametes and blockade of fertilization in vitro and in vivo

Two individual assays with gametes, immunofluorescent staining with mAbs and ookinete conversion, were carried out in vitro. The change of environment from the bloodstream to the mosquito midgut triggers the development of gametocytes in infected erythrocytes into mature, highly motile 'male' microgametes and more sessile 'female' macrogametes and gamete emergence from erythrocytes. These events can be mimicked in vitro by reducing the temperature or pH and adding xanthurenic acid to the medium (ookinete medium). Each male gametocyte gives rise to eight microgametes, which look like a number of waving cilia as they emerge from the infected erythrocyte; this process is thus termed 'exflagellation'.

Reactivity of mAbs to native HAP2 in *P. berghei* microgametes was determined using cultured, infected erythrocytes undergoing exflagellation. After cultures were fixed, gametes adsorbed to slides were incubated with mouse D3 mAb and rabbit anti-alpha-tubulin followed by staining with fluorochrome-conjugated secondary antibodies specific for mouse and rabbit IgG and DAPI.

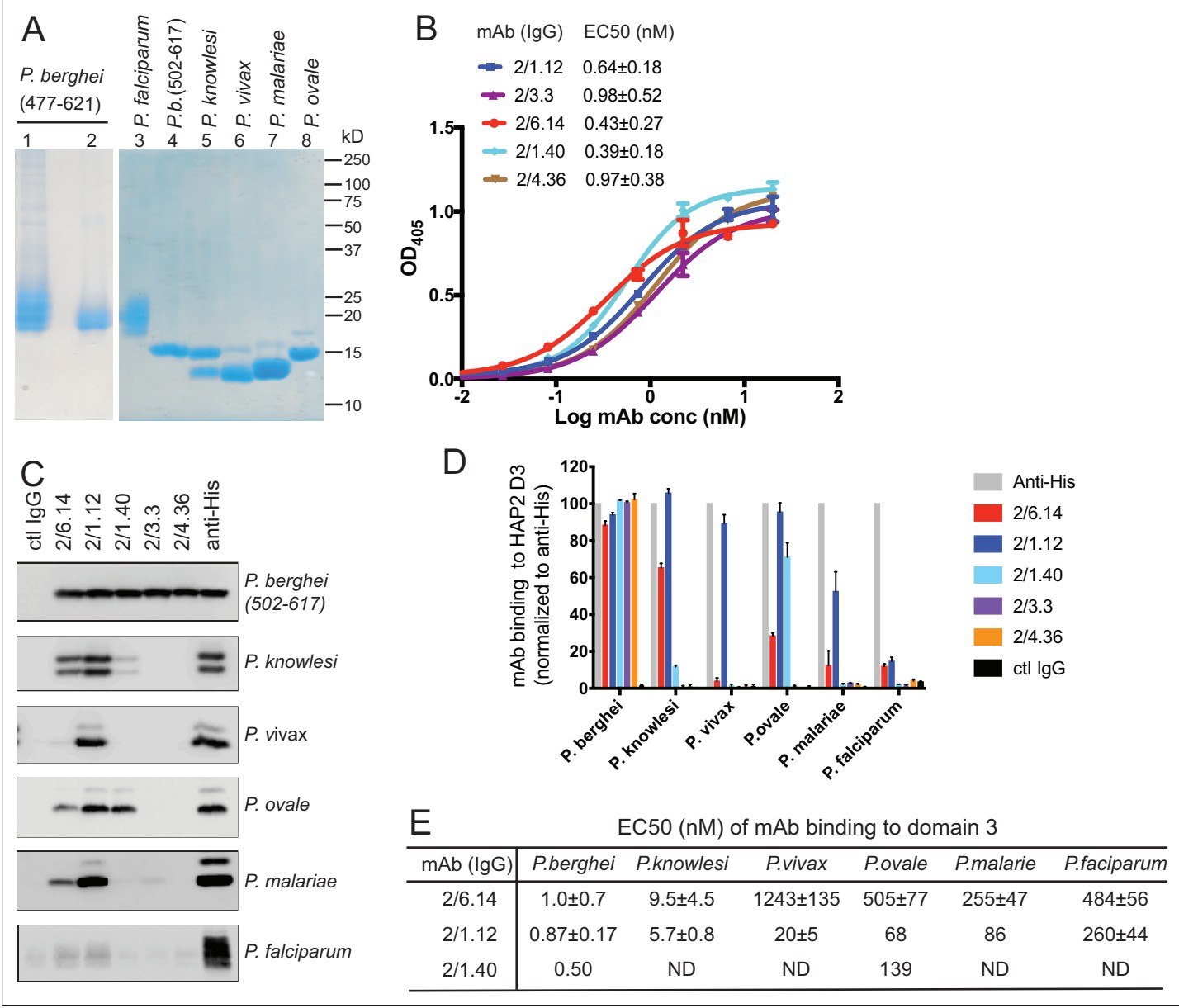

**Figure 2.** Cross-reactivity of HAP2 D3 mAbs among *Plasmodium* species. (**A**) Reducing SDS 12.5% PAGE of purified HAP2 D3 stained with Coomassie blue. Lanes 1 and 2: *P. berghei* HAP2 D3 (aa 477–621) purified by Ni-NTA (lane 1) and then treated with Endo D and purified by gel filtration (lane 2). Lanes 3–8: Purified HAP2 D3 fragments from *Plasmodium* species treated with Endo D or from *P. berghei* (aa 502–617) with N-linked sites removed by mutation. (**B**) Titration of antibody binding to Endo D-treated *P. berghei* HAP2 D3 (aa 477–621) by ELISA. Sigmoidal curve fits show one representative experiment with mean ± difference from the mean of duplicates; EC50 values show mean ± SD of three experiments. (**C**) Cross-species reactivity of *HAP2* D3 mAbs by immunoprecipitation. Purified D3 proteins shown in (**A**) (2 µg each) were subjected to immunoprecipitation with the indicated mAbs, anti-His 1/5.13, or control IgG (8 µg each). Immunocomplexes were analyzed by reducing 12.5% SDS-PAGE and Western blot using rabbit polyclonal antibodies to the C-terminal His tag of D3. (**D**) Quantitation of results from (**C**) and a repeat experiment. Intensities of bands were quantitated and data normalized to anti-His. Results shown are averages of the two experiments ± difference from the mean. (**E**) EC50 measurements of HAP2 D3 binding to immobilized mAbs by ELISA. Purified D3 proteins shown in (**A**), lanes 3–8, were used. Sigmoidal curve fitting of D3 titration (*Figure 2—figure supplement 3*) and EC50 used GraphPad Prism 7 software. EC50 shown are mean ± difference from means of two experiments, each experiment with triplicates. ND, Not determined.

The online version of this article includes the following figure supplement(s) for figure 2:

**Figure supplement 1.** Surface plasmon resonance (SPR) analysis of binding Interactions of Fab 2/6.14 and Fab 2/1.12 with PbHAP2 D3 and monomeric ectodomain.

**Figure supplement 2.** Reactivity of mAb 1/5.13 to fusion proteins with His tags.

**Figure supplement 3.** Titration of binding of HAP2 D3 from *Plasmodium* species to immobilized mAbs.

Anti-tubulin and DAPI were used to identify microgametes by their highly elongated microtubule cytoskeletons and nuclei. All 5 D3 mAbs specifically stained *P. berghei* microgametes (*Figure 3A*).

We next tested the ability of the D3 mAbs to inhibit *P. berghei* microgametes from fertilizing macrogametes and forming ookinetes in vitro. Mouse blood infected with *P. berghei* gametocytes was mixed with D3 mAbs or control mouse IgG diluted in ookinete medium. After 24 hr, ookinetes and unfertilized female gametocytes were identified by their morphology and staining with a fluorescent antibody, counted, and ookinete conversion rates were calculated. Only a single microgamete is required to fertilize a macrogamete, and each ookinete observed comprises a successful fertilization event. In control IgG, conversion rates were ~80% (*Figure 3B*). Strikingly, mAb 2/6.14 at 500 µg/ml completely inhibited ookinete conversion and at 250 µg/ml inhibited by 96% (p < 0.001). In contrast, none of the other mAbs inhibited by >50%, although mAb 2/1.40 showed significant inhibition at 500 µg/ml but not at 250 µg/ml (*Figure 3B*).

We then tested mAb 2/6.14 for its ability to block transmission in vivo in mosquitoes. Female *Anopheles stephensi* mosquitoes were allowed to feed through membranes on blood from *P. berghei* infected mice mixed with mAb 2/6.14 or control IgG. After 14 days, mosquito midguts were dissected and oocysts/midgut were counted, that is, oocyst intensity was determined. Oocyst intensity was significantly reduced by mAb 2/6.14, particularly at 250 and 100 µg/ml (*Figure 3C*). We also measured the prevalence of infection among the fed mosquitoes, i.e. the % of mosquitoes with at least one oocyst. Prevalence was significantly reduced by mAb 2/6.14 at both 250 and 100 µg/ml (*Figure 3— figure supplement 1*). Summing the results over all mosquitos in the three independent experiments showed significant inhibition of intensity at all three mAb 2/6.14 concentrations, significant inhibition of prevalence at 250 and 100 µg/ml of mAb 2/6.14, dose-dependent reduction of both measures over the three antibody concentrations, and a reduction of intensity by 85% at 250 µg/ml (*Figure 3D*). Overall, the results show that all five antibodies react with microgametes, that mAb 2/6.14 exhibits potent ability to block *P. berghei* fertilization in vitro and in vivo, and thus that purified HAP2 D3 has potential as a transmission-blocking vaccine.

## Structural characterization of *P. berghei* HAP2 and complexes of D3 with Fabs

To obtain structural insights relevant to rational development of a transmission-blocking vaccine, we obtained crystal structures of the P. *berghei* HAP2 D3 502–617 fragment with its three N-linked sites mutated out in complex with Fab fragments. Diffraction data was collected and refined to resolutions of 2.8 and 2.1 Å with Rfree of 29% and 23% for complexes with the transmission-blocking Fab 2/6.14 and cross-reactive Fab 2/1.12, respectively (*Table 1*). Each structure has two independent D3-Fab complexes in the asymmetric unit, giving us four examples of D3. The two Fab fragments bind to opposite faces of D3 (*Figure 4A and B*).

D3 contains seven β-strands labeled A to E that are arranged into two β-sheets containing the ABE and DCFG β-strands (*Figure 4B and C*). These β-sheets associate through hydrophobic faces to form a β-sandwich. The way in which the sequence folds into this arrangement classifies it as a fibronectin type III (Fn3) domain. The D3 β-strands connect to one another through loops at each end of the domain. The A-B, C-D, and F-G loops link adjacent β-strands within a sheet and the B-C, D-E, and E-F loops link the two sheets.

Comparisons among the four examples of D3 show markedly different conformations for the loops at the C-terminal end of D3, which abut the polypeptide segment that connects to the plasma membrane (*Figure 4C*). Quantitation of flexibility in these loops, A-B, C-D, and E-F, shows high root mean square deviation (RMSD) values or lack of comparison because of residues missing in density (dashes, *Figure 4F*). In contrast, the loops at the N-terminal end that abut D1 show very similar conformations in all independent crystallographic examples. These loops, B-C, D-E, and F-G, show low RMSD values (*Figure 4F*), consistent with the presence of several backbone-backbone or sidechain-backbone hydrogen bonds that stabilize each loop. Three sidechains that support the hydrogen bond networks in these loops, Asn-531 and Asn-532 in the B-C loop and Asp-598 in the F-G loop (overlined in *Figure 4F*), are invariant in the 6 *Plasmodia* species.

The two antibodies bind largely to opposite faces of D3, with 2/1.12 binding primarily to the ABE sheet and 2/6.14 binding primarily to the DCFG sheet (Fab contacts are colored in green and red, respectively in *Figure 4B*). The overall shapes of the contact surfaces are shown in open book views in

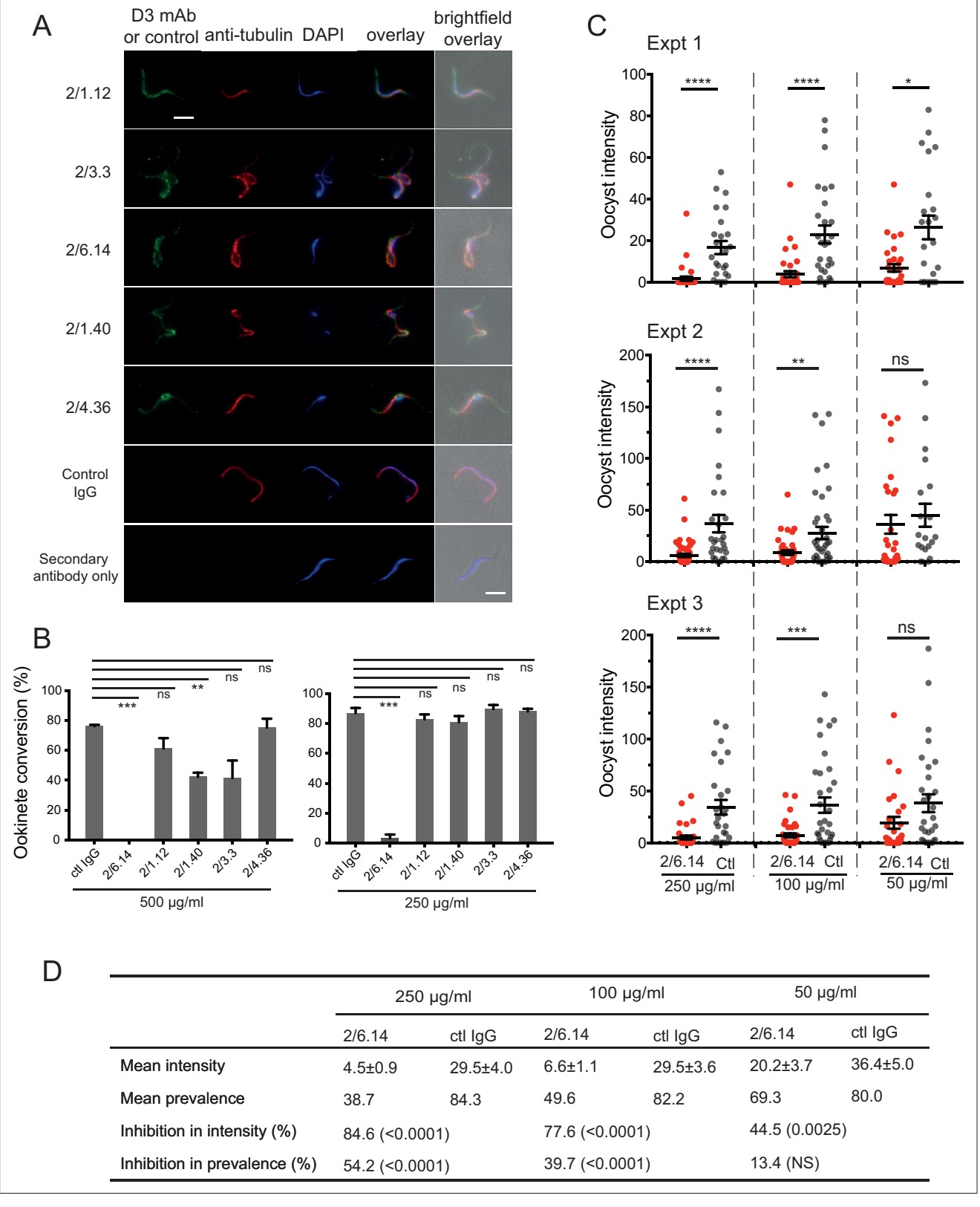

**Figure 3.** Immunofluorescent staining and inhibition of *P. berghei* fertilizaton by mAbs. (**A**) Immunofluorescent staining of microgametes from *P. berghei*. Fixed microgametes were stained with mouse D3 mAbs or control IgG, rabbit anti-α-tubulin, secondary Alexa Fluor-488 anti-mouse IgG and Alexa Fluor-594 anti-rabbit IgG, DAPI, and imaged with epifluorescence. Scale bars = 5 μm. (**B**) Fertilization in vitro measured as macrogamete conversion to ookinetes (% ookinetes/(ookinetes + macrogametes)) in the presence of indicated antibodies. Results are mean ± SEM of three

*Figure 3 continued on next page*

*Figure 3 continued*

independent experiments, analyzed by paired t test: *** p < 0.001, **p < 0.01, ns, non-significant p > 0.05. Total number of macrogametes + ookinetes in all three experiments are from right to left at 500 µg/ml: 139, 177, 147,104, 199 and 129, and at 250 µg/ml: 222, 120, 270, 410, 275, and 306. (**C**) ( **and D**) Mosquitoes were allowed to feed on antibody or control IgG diluted in infected mouse blood placed in membrane feeders; results are shown from three independent experiments. (**C**) Oocyst intensities (oocysts per mosquito) are shown as filled circles with means as horizontal bars ± SEM. ****p < 0.0001, ***p = 0.0001–0.001, **p = 0.001–0.01, *p = 0.01–0.05, ns p > 0.05 by Mann-Whitney test. (**D**) Summary of intensity and prevalence (% of infected mosquitoes) with SEM from all three experiments. Prevalence and N values are shown in *Figure 3—figure supplement 1*. Inhibition (%) was calculated relative to negative control IgG at the same concentrations. The significance of inhibition is shown in parentheses for intensity (Mann-Whitney test) and prevalence (Fisher's exact test).

The online version of this article includes the following figure supplement(s) for figure 3:

**Figure supplement 1.** Transmission blocking activity of mAb 2/6.14 in standard membrane feeding assay.

which the Fab and D3 are rotated apart like two facing pages in an open book (*Figure 4D and E*). In each Fab, both the heavy and light chain variable domains contribute significantly to the contact with D3. The sidechains with major contacts ( > 10 Å$^2$ of buried solvent accessible surface area or H-bonds) are shown as large colored circles over the alignment in *Figure 4F* and shown in stick in *Figure 4G and H*.

Although the 2/1.12 Fab primarily engages the ABE β-sheet, it extends over the edge of this sheet to also contact the D strand, where it overlaps with the 2/6.14 epitope (*Figure 4B and F*). Fab 2/1.12 has the most contact with four residues in β-strands A, B, and E of D3 (*Figure 4G*). The sidechains of Thr-508 from D3 β-strand A and His-528 from strand B form hydrogen bonds with the sidechain of Tyr-32 from light chain complementarity-determining region 1 (L-CDR1). In adjacent interactions, Tyr-577 in β-strand E forms a sidechain hydrogen bond to the backbone of Gly-91 in L-CDR3 while the Tyr-577 backbone hydrogen bonds to the sidechain of Asn-104 in H-CDR3. Another close contact and backbone-backbone hydrogen bond between D3 E β-strand residue Ala-575 and H-CDR3 residue Tyr-102 further strengthens interaction (*Figure 4G*). All four of these major epitope residues are invariant among the species examined here (*Figure 4F*), correlating with excellent cross-reactivity of mAb 2/1.12.

Although mAb 2/6.14 contacts residues in each β-strand of the D3 DCFG β-sheet, the major contacts are formed by Arg-565 in strand D and six residues in the F-G loop (*Figure 4F and H*). The F-G loop forms a large bulge on the D3 surface that fits into a cavity in the Fab centered between the three H chain CDR loops (*Figure 4E*). The Arg-565 sidechain forms bidentate hydrogen bonds to the backbone of Tyr-91 in L-CDR3 and a cation-pi interaction with Tyr-32 in L-CDR1. All three Fab H chain CDR loops surround the D3 F-G loop with each CDR forming at least one of a total of four mainchain-mainchain hydrogen bonds. A network of two sidechain-sidechain and two sidechain-backbone hydrogen bonds forms around FG loop residue Glu-603 and H-CDR2 residue Asn-56. Hydrogen bonds between Lys-597 and Ser-57 of H-CDR2 and salt bridge between Lys-602 and Asp-31 of H-CDR1 further extend the interaction with the F-G loop (*Figure 4H*).

D3 contains six cysteine residues, all of which are conserved among *Plasmodium* species; however, only four of these form disulfide bonds (*Figures 4F and 5A*). Cys-513 in the A-B loop forms a short-range disulfide to Cys-523 in the B strand that likely stabilizes the residues between Cys-513 and the beginning of the B strand, which are highly polar and mostly disordered. Cys-546 in the C strand and Cys-592 in the F strand form a long-range disulfide bond to bridge these two adjacent β-strands in the middle of the DCFG β-sheet. Two D3 cysteines are free. Free cysteines buried in hydrophobic regions are occasionally present in extracellular proteins. Cys-542 is well buried in the hydrophobic core. Cys-604 is exposed in D3-Fab complexes but in the full ectodomain may be buried by residues at the interface between D1 and D3, including residues N-terminal to D3 residue 502, which were not present in the crystallization construct (*Figure 5A and B*).

HAP2 D3 in *P. berghei* is only 9% and 18% identical in sequence to *C. reinhardtii* and *A. thaliana* HAP2 D3, respectively; however, their three-dimensional structures are highly conserved (*Figure 5B*). Their seven β-strands superimpose with very low RMSD (*Figure 5C*). The long-range disulfide bond that connects the C and F strands is also conserved (*Figure 5C*). Both *C. reinhardtii* and *P. berghei* but not *A. thaliana* have long insertions in their A-B loops and a disulfide bond that may help to stabilize these long loops, which orient quite differently (*Figure 5B,C*).

**Table 1.** Statistics of X-ray diffraction and structure refinement of *Pb*HAP2 domain 3 (D3) complexed with 2/6.14 Fab or 2/1.12 Fab.

| | D3-2/6.14 Fab | D3-2/1.12 Fab |
|---|---|---|
| **Data collection statistics** | | |
| Space group | P222 | P2$_1$ |
| α, β, γ, ° | 90, 90, 90 | 90, 95.8, 90 |
| Unit cell (a, b, c), Å | 78.3, 122.6, 168.5 | 43.9, 187.1, 74.4 |
| Resolution range (Å) | 50.0–2.80 (2.87-2.80)* | 50.0–2.10 (2.15–2.10) |
| Completeness (%) | 99.3 (99.5) | 98.4 (96.9) |
| Number unique reflections | 40,741 (2,938) | 68,138 (4,935) |
| Redundancy | 3.8 (4.0) | 3.6 (3.7) |
| R$_{merge}$ (%)[†] | 5.7 (70.2) | 8.3 (153) |
| I/σ(I) | 12.8 (2.2) | 8.42 (0.52) |
| CC$_{½}$ (%) [‡] | 99.8 (90.9) | 99.7 (10.9) |
| Wavelength (Å) | 1.0332 | 1.0332 |
| **Refinement statistics** | | |
| R$_{work}$ (%) [§] | 25.3 (33.7) | 18.97 (37.3) |
| R$_{free}$ (%) | 29.2 (34.8) | 23.27 (40.5) |
| Bond RMSD (Å) | 0.003 | 0.003 |
| Angle RMSD (°) | 0.548 | 0.603 |
| Ramachandran plot [¶] (Favored/allowed/outlier) | 91.5/7.6/0.9 | 95.93/4.07/0.1 |
| **Number of atoms** | | |
| Protein | 10,310 | 8,183 |
| Ligand | 22 | 54 |
| Water | 18 | 353 |
| **B factor** | | |
| Protein | 128.6 | 72.7 |
| Ligand | 112.8 | 112.9 |
| Water | 138.0 | 64.9 |
| **Molprobity percentile** | | |
| (Clash/Geometry) | 98/97 | 99/97 |
| PDB | 7LR3 | 7LR4 |

* The numbers in parentheses refer to the highest resolution shell.
[†] R$_{merge}$ = Σ h Σ i |Ii(h) -< I(h)> | / Σ h Σ i Ii(h), where Ii(h) and< I(h)> are the i$^{th}$ and mean measurement of the intensity of reflection h.
[‡] Pearson's correlation coefficient between average intensities of random half-datasets for each unique reflection (**Karplus and Diederichs, 2012**).
[§] R$_{factor}$ = Σ h||Fobs (h)|-|Fcalc (h)|| / Σ h|Fobs (h)|, where Fobs (h) and F calc (h) are the observed and calculated structure factors, respectively. No I/σ(I) cutoff was applied.
[¶] Calculated with MolProbity (**Davis et al., 2007**).

There is little variation in the orientation and length of the BC, DE and FG loops at the N-terminal end of D3. These loops are stabilized by an Asn in the BC loop, an Asp in the FG loop, and the hydrogen bonds they make. Asn-531 in *P. berghei* (asterisked in *Figure 5C*), forms hydrogen bonds to both the BC and DE loops near the D1 junction (*Figure 5D and E*). Invariance in the other two species, despite the evolutionary distance between the unicellular algae, the plant, and protozoan parasite, is of particular interest because Asn-531 is in an N-glycosylation NX(S/T) sequon in *P. berghei* but its equivalents in the other species are not (*Figure 5C*). The two other species are known to be N-glycosylated but *Plasmodium* species are not. In D3 constructs mutated to remove N-linked sites, the N was left unchanged because it was much more conserved than the (S/T) position in other *Plasmodium* species. The B, D, and E strands that connect in these loops all contribute residues important in the mAb 2/1.12 epitope (*Figure 5E*).

To better understand the *P. berghei* HAP2 ectodomain and its binding to 2/6.14 Fab, we obtained negative stain EM images and subjected ~2000 particles to iterative alignment and classification. The *P. berghei* HAP2 ectodomain is rod-like with three or four oval densities arranged linearly along the rod (*Figure 4I* panel 1 and *Figure 4—figure supplement 1*). The 2/6.14 Fab - ectodomain complex is L-shaped (*Figure 4I* panel 2). Similar results were obtained for complexes with ectodomains with residues 43–617 and 61–611 (*Figure 4—figure supplement 1*). The two globules with stronger density correspond to the Fab, which has two domains per globule. The globules with weaker density correspond to the HAP2 ectodomain. Fab 2/6.14 binds essentially perpendicularly to one end of the HAP2 rod, in agreement with the crystal structure showing that the Fab binds to the side of D3 (*Figure 4A*). The crystal structure of the 2/6.14 Fab-D3 complex cross-correlated well with the entire ectodomain Fab complex (*Figure 4I* panel 3), and the ribbon cartoon in the same orientation had its N-terminus facing D1 as expected (*Figure 4I* panel 4).

## Differences in antibody reactivity with D3 and the ectodomain

Because of their differences in blocking transmission, we wondered whether mAbs differed in reactivity with the HAP2 ectodomain. Yields of monomeric ectodomain were lower than for D3 as described in Methods; nonetheless, gel filtration yielded a sharp peak (*Figure 6A*). The peak

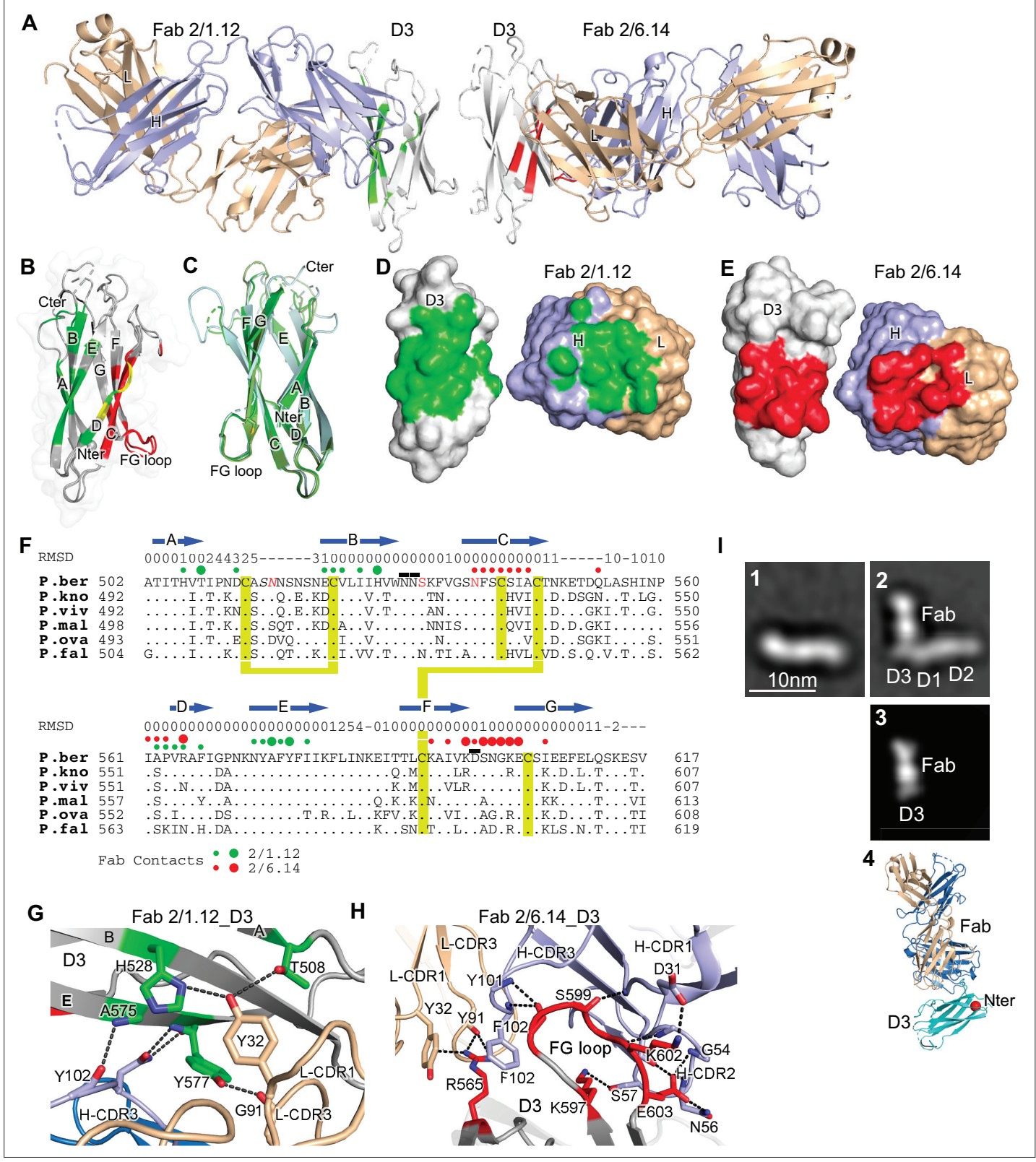

**Figure 4.** Structures of HAP2 D3 in complex with 2/6.14 Fab or 2/1.12 Fab. (**A**) Overview of D3-2/1.12 Fab (left) and D3-2/6.14 Fab complexes (right). (**B**) Cartoon representation of D3 with residues colored that contact Fab 2/1.12 (green), Fab 2/6.14 (red) or both Fabs (yellow). (**C**) Cartoon representations of three independent molecules in the two crystals, excluding the one molecule with multiple conformations. Each molecule is colored in a different variation of green. Dashes symbolize loops with missing density. (**D**) and (**E**) Solvent accessible surfaces of complexes are shown in open book views

*Figure 4 continued on next page*

*Figure 4 continued*

in which the Fab and D3 components are rotated apart like two facing pages in an open book. In B, D and E, atoms in D3 and Fabs within 5 Å of one another are colored green (Fab 2/1.12 complex) or red (Fab 2/6.14 complex) and are otherwise white (D3), light blue (Fab H chain), or wheat (Fab L chain). (F) Structural and sequence conservation of D3 and its 2/1.12 and 2/6.14 epitopes. Top line: β-strands shown as arrows. Second line: RMSD of Cα atom position (Å) among the two independent Fab 2/1.12 - D3 complexes and the Fab 2/6.14-D3 complex with a single conformation, calculated after structure alignment by RaptorX (*Wang et al., 2013*). A dash shows positions where residues were defined in only 0 or 1 of the three structures. Third line: Filled circles show Fab contacts (green for 2/1.12 Fab and red for Fab 2/6.14); residues that mediate major interactions ( > 10 Å² of buried solvent accessible surface area or H-bonds) or minor interactions ( < 10 Å² buried solvent accessible surface) are shown with large or small circles, respectively. Remaining lines show D3 sequence in *P. berghei*, *P. knowlesi*, *P. vivax*, *P. malariae*, *P. ovale*, and *P. falciparum* with identities to *P. berghei* shown as dots. *P. berghei* residues in red are wild-type and were mutated to remove N-linked sites. Residues in italics were not visualized in any of the three structures. *P. berghei* residues with sidechains mentioned in the text that stabilize hydrogen bonds in the B-C and F-G loops are overlined. Cysteines are highlighted in yellow and connected when disulfide-linked. (G) and (H). Details of D3 interactions with the 2/1.12 Fab (G) and 2/6.14 Fab (H). D3 is silver, Fab H and L chains are light blue and wheat, respectively, and residues with major interactions with Fabs as defined in (F) have carbons colored green (G) or red (H). Dashes show hydrogen bond and pi-cation interactions. (I) The most populated negative stain EM class averages of the HAP2 ectodomain (residues 43–617) alone (panel 1, 1,128 particles) and in complex with the 2/6.14 Fab (panel 2, 379 particles). Panel 3 shows the best correlating projection from the complex crystal structure. Panel 4 shows a ribbon diagram of the crystal structure in the same orientation. D3 colored cyan, Fab H and L chains colored light blue and wheat, respectively, and N-terminal of D3 shown as a red sphere.

The online version of this article includes the following figure supplement(s) for figure 4:

**Figure supplement 1.** EM class averages.

**Figure supplement 2.** Steric clashes of mAbs 2/6.14 and 2/1.12 with the postfusion HAP2 conformation.

primarily contained a band at 75 kDa, in agreement with the expected size of the ectodomain; furthermore, its C-terminus was present as shown by detection of the C-terminal His-tag by western blotting (*Figure 6B*). Treatment with Endo D reduced the size of the ectodomain to ~71 kDa (*Figure 6B*).

Presence of D3 epitopes in the ectodomain was tested by immunoprecipitation. Immunoprecipitation of the ectodomain, and D3 as a control, was detected using western blotting of the His tag (*Figure 6C*). mAb 2/6.14 was fully reactive with the ectodomain as shown by immunoprecipitation comparable to that with the His-tag antibody. In contrast and surprisingly, the other four mAbs were only partially reactive with the ectodomain, although they reacted with D3 comparably to mAb 2/6.14.

These differences were followed up by further comparisons that focused on mAbs 2/6.14 and 2/1.12. Calculations of affinity and dissociation constants assume fully active material; if only a fraction is active then the apparent affinity is lower. In SPR, only the concentration of the analyte enters into affinity calculations. Therefore, we compared affinities measured for the two Fabs using them either as analyte or immobilized on the chip (*Figure 6E* and *Figure 2—figure supplement 1*). For Fab 2/1.12, the ratio of $K_D$ values for D3 (analyte/immobile) was 5.2/5.9 = 0.88 ± 0.09 (*Figure 6E*). In contrast, these values for the ectodomain were 106/11.7 = 9.1 ± 6.2. Thus, the ectodomain appears to be only 0.88/9.1 = 9% ± 6% active in binding to Fab 2/1.12. Comparable $K_D$ value ratios for Fab 2/6.14 were 6.0/10.2 = 0.59 ± 0.23 for D3 and 22.9/50.5 = 0.45 ± 0.04 for the ectodomain. Thus, the ectodomain appears to be 0.59/0.45 = 130% ± 52% active in binding mAb 2/6.14, within error of the expected value of 100%, while Fab 2/1.12 binding to the ectodomain was far below the expected value.

We also assessed epitope content of the ectodomain by mixing it with equimolar concentrations of Fab and separating the complexes from components in gel filtration. Gel filtration of the mixtures was compared to that of the same concentrations of unmixed components (*Figure 6D*). With Fab 2/6.14, much of the ectodomain was shifted to higher molecular weight and the Fab fragment was substantially depleted. With Fab 2/1.12 in contrast, only a small portion of the ectodomain was shifted to a higher molecular weight shoulder and there was little or no depletion of the Fab.

We further measured affinity with bio-layer interferometry (BLI) and complex formation with an independent preparation of the ectodomain. mAb IgG were immobilized on anti-mouse Fc capture sensors. HAP2 D3 bound well to both mAbs 2/6.14 and 2/1.12. In contrast, HAP2 ectodomain bound to mAb 2/6.14 but not to mAb 2/1.12 (*Figure 6—figure supplement 1*). In conclusion, mAb 2/6.14 binds well to both D3 and the ectodomain, whereas mAb 2/1.12 and three other antibodies bind well to the D3 immunogen but bind poorly to or do not recognize the ectodomain.

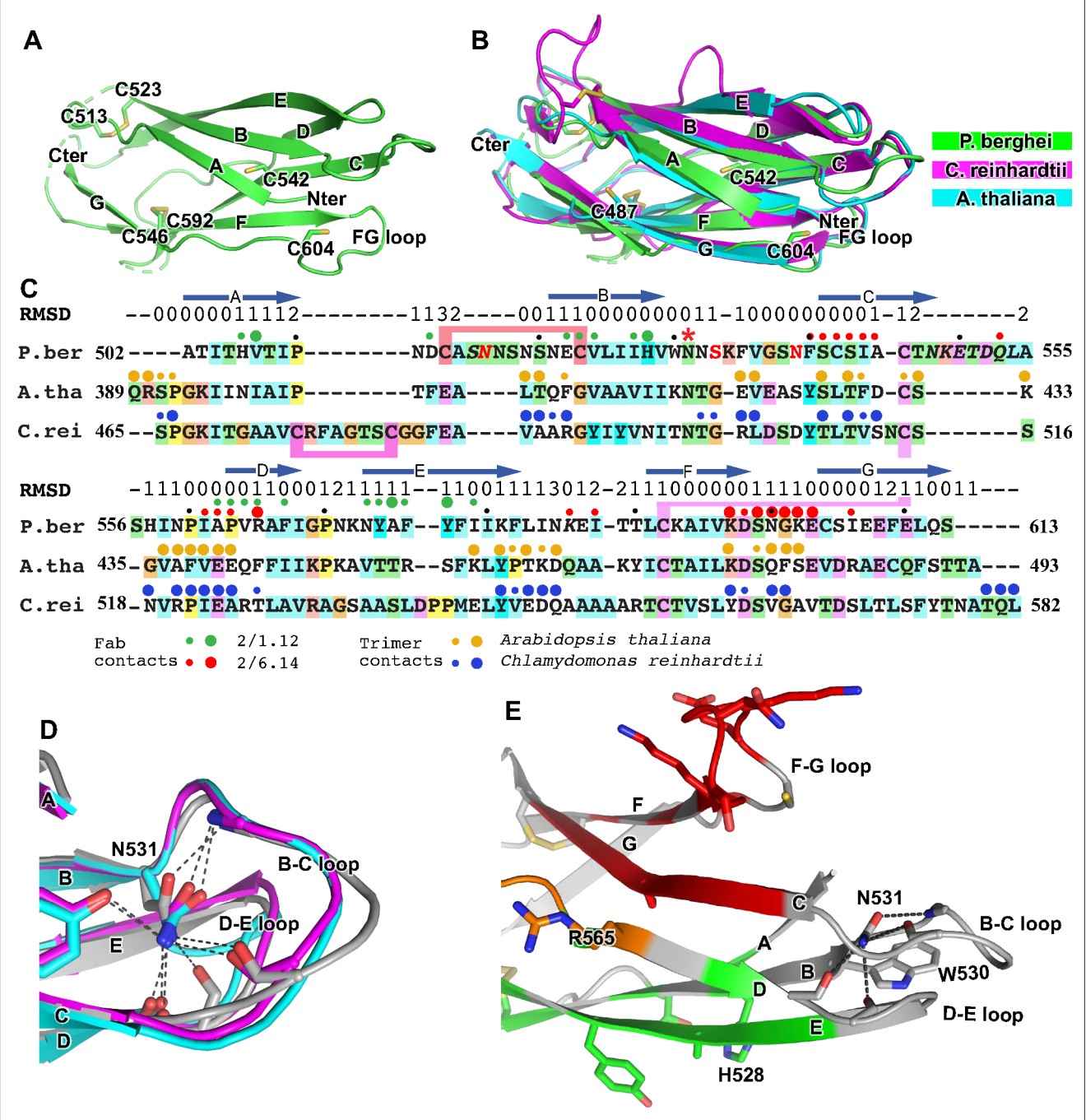

**Figure 5.** Structural conservation of D3 of HAP2 across phyla. (**A**) *P. berghei* HAP2 D3 from the Fab 2/1.12 complex showing the disulfide-bonded and free cysteines. (**B**) Superimposition of HAP2 D3 from the *P. berghei* 2/6.14 Fab complex and trimeric fusion states of *Chlamydomonas reinhardtii* (pdb ID:6DBS) and *Arabidopsis thaliana* (pdb ID: 5ow3). Structures were aligned using RaptorX (***Wang et al., 2013***). (**C**) Structure-based sequence alignment from the superimposition shown in (**B**). β-strands and RMSD (Å) of Cα atom positions are shown above the sequences. Green and red filled circles above the *P. berghei* sequence show Fab 2/1.12 and Fab 2/6.14 contacts, respectively as defined in ***Figure 4F*** legend. In the *P. berghei* sequence, the three residues in red mark residues that were mutated to remove N-linked sites. Orange and blue filled large and small circles above *Arabidopsis thaliana* and *Chlamydomonas reinhardtii* sequences mark residues buried in trimer contacts with >10 Å² burial or a hydrogen bond or <10 Å², respectively. Solvent accessible surface area burial was calculated with PISA (***Krissinel and Henrick, 2007***). (**D**) Asn-531, present in an N-glycosylation sequon in the *P. berghei* sequence, forms stabilizing hydrogen bonds to the backbones of the B-C and D-E loops, a function that is conserved in HAP2 in other phyla. The color code is the same as in panel B, except *P. berghei* is in silver. (**E**) Asn-531 locates near mAb epitopes. Asn-531 and sidechains or backbones to which it hydrogen bonds are shown in stick. Residues with major or minor contacts with Fabs as defined in ***Figure 4F*** legend are shown with both backbone and sidechain, or backbone only, respectively, and colored according to the ***Figure 4F*** legend and the key. Residues in both epitopes are in orange.

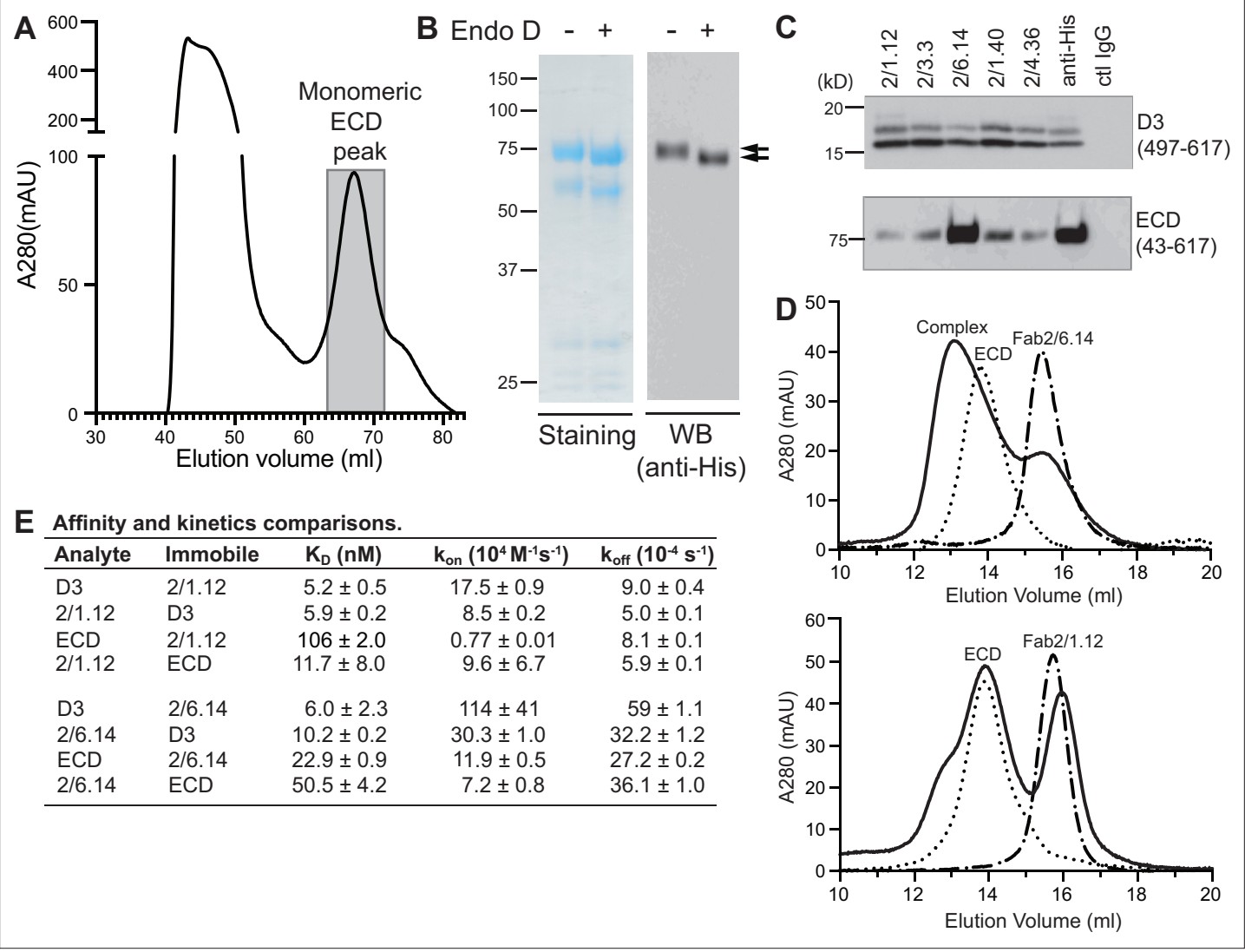

**Figure 6.** Antibody binding to the HAP2 ectodomain. (**A**) Gel filtration profile of Ni-NTA affinity-purified ectodomain (residues 43–617) on a Hiload 16/600 Superdex 200 column. (**B**) Reducing SDS 7.5% PAGE and Coomassie blue staining and Western blot (WB) of the peak eluted at 67.2 mL (shaded in A). Aliquots were treated with or without Endo D to remove N-linked glycans. Arrows indicate ectodomain bands before and after Endo D treatment. The HAP2 ectodomain contains 11 putative N-glycosylation sites; however, Endo D treatment reduced the mass by only ~3 kD. It is unknown how many sites are N-glycosylated and whether all sites are accessible to Endo D. (**C**) Immunoprecipitation. Reactivity of D3 mAbs to HAP2 ectodomain (bottom) in comparison to D3 (top). Two µg purified ectodomain from (**A**), without Endo D treatment, was subjected to immunoprecipitation with the indicated D3 mAbs, anti-His mAb 1/5.13 or control IgG. D3 (497–617, containing three putative N-glycosylation sites) was immunoprecipitated from culture supernatants of S2 transfectants. Multiple D3 bands are different glycoforms. (**D**) Analysis of formation of PbHAP2 ectodomain complex with 2/6.14 Fab (top) and 2/1.12 Fab (bottom) by gel filtration. Ectodomain and Fab 2/6.14 or Fab 2/1.12 were mixed in 20 mM Tris-HCl pH 8.0, 500 mM NaCl at 1:1 molar ratios and incubated at 4 °C for 1 hr. Mixtures was then subjected to gel filtration on a Superdex 200 10/300 GL column in the same buffer. Elution profiles of the mixture, ectodomain or Fabs are shown as solid, dotted, or dashed lines, respectively. (**E**) $K_D$ values ($k_{off}/k_{on}$) and kinetic values measured by surface plasmon resonance. Values are mean ± difference of the means from two independent experiments. D3 contained residues 502–617 with N516T, S533N and N539Q mutations and the ectodomain (ECD) contained residues 43–617 without Endo D treatment.

The online version of this article includes the following figure supplement(s) for figure 6:

**Figure supplement 1.** Biolayer interferometry (BLI) analysis of binding Interactions of PbHAP2 D3 and monomeric ectodomain with IgG 2/6.14 and IgG 2/1.12.

## Discussion

We have generated a mAb to HAP2 D3 that blocks *Plasmodium* ookinete formation in vitro and in vivo and have determined a crystal structure of its complex with D3 that reveals the epitope to which the neutralizing mAb binds. Furthermore, we found that a subset of D3 mAbs were not fully reactive with the HAP2 ectodomain and that these antibodies were ineffective in preventing ookinete conversion. These findings have important implications for choice of future HAP2 immunogens. Previously, in the absence of structural knowledge, a large number of *P. berghei* HAP2 cDNA clones beginning and ending in different positions were tested in *E. coli*. Only one sequence, residues 355–609, was found to be expressed well; fortunately, it yielded an immunogen that elicited polyclonal antisera that inhibited gamete fertilization in vitro and reduced transmission in vivo (*Blagborough and Sinden, 2009*). This sequence contained about half of D2, a small portion of D1, and all D3 except for the last four residues of strand G. Recently, polyclonal antibodies to the putative fusion loops of D2 of *Plasmodium berghei* HAP2 also showed efficacy in blocking fertilization (*Angrisano et al., 2017*). The same laboratory and procedures were used for these studies, so the results are directly comparable, except for the use of a polyclonal antibody in the previous and monoclonal antibody in the current study. Surprisingly, mAb 2/6.14 to D3 is at least fivefold more effective than that affinity-purified polyclonal antibody, as 100 µg/ml reduced oocyst intensity by 78%, whereas 500 µg/ml affinity-purified polyclonal antibodies to the fusion loop reduced intensity by 61%. Recently, residues 231–459 of *P. vivax* HAP2 were also tested. It contains ~30% of D1 and ~60% of D2 and, as expected from lack of a complete domain, had to be recovered from inclusion bodies in insect cells (*Qiu et al., 2020*). This material elicited antibodies that inhibited transmission, but the results cannot be directly compared. The present study is the first time that a defined domain from HAP2 has been used in immunization, that a monoclonal antibody has been found to be effective in blocking transmission, and that a *Plasmodium* species HAP2 structure has been reported. Thereby, this study advances the rational development of transmission-blocking malaria vaccines.

HAP2 is highly conserved, both within and between *Plasmodium* species. In *P. falciparum*, 199 isolates in PlasmoDB (https://www.plasmodb.org, release 54) evaluated have identical D3 amino acid sequences and only a few polymorphisms are present in D1 and D2 of HAP2 ectodomain in two or more isolates (R99E, n = 2; N184S, n = 14; D185Y/E, n = 11/7 and D455N, n = 47). The high-sequence identity of gamete antigens contrasts to malaria vaccine antigens expressed by sporozoites such as TRAP and the C-terminal region of CSP used in the RTS,S vaccine and those expressed by blood stage parasites such as MSP1 and AMA1, which have high levels of polymorphisms and have proven to be challenging for vaccine antigen design (*Neafsey et al., 2015*; *Takala et al., 2009*; *Barry and Arnott, 2014*; *Ouattara et al., 2015*). Compared to CSP and TRAP, HAP2 is also highly conserved among species; for example, *P. berghei* HAP2 is 60–70% identical in D3 to the five species we examined that are capable of infecting humans.

We structurally characterized HAP2 D3 in complex with two Fab fragments that bound to opposite faces of D3, on the β-sheets that form its β-sandwich domain. Conservation or variation among residues in these interfaces among *Plasmodium* species provides insights into which residues are important in their epitopes. The four residues most important for mAb 2/1.12 binding were identical among all six species examined, explaining the wide cross-reactivity of this mAb. Although the D3 F-G loop was central in the 2/6.14 epitope, many of its contacts were between backbone atoms. The sidechain with the most contacts was Arg-565 in the D strand. All three interactions by the Arg-565 sidechain would be abolished by the Asn substitution in *P. vivax* and *P. falciparum* (*Figure 4F*), which had the lowest affinity.

Thus far, structural information on HAP2 has been on its post-fusion, trimeric form or on isolated domains (*Fédry et al., 2017*; *Feng et al., 2018*; *Fedry et al., 2018*; *Baquero et al., 2019*). The rod-like, linear conformation of the HAP2 ectodomain and its monomeric state has not previously been seen for HAP2. Its resemblance to the linear conformation of pre-fusion states of structurally homologous class II viral fusogens suggests that this linear conformation of the HAP2 ectodomain corresponds to its pre-fusion state. The pre-fusion state is preferred over the post-fusion state as the target for transmission blocking antibodies, because antibodies that bind to the pre-fusion state not only can act earlier but can also sterically block the interfaces required for trimer formation and the fold-back of D3 onto D1 and D2 (*Figure 1*).

In addition to immunological mechanisms for blocking gamete fertilization in the mosquito blood meal (*Graves et al., 1985*), mAb 2/6.14 might also block conformational changes required for conversion of the monomeric pre-fusion state of HAP2 to the trimeric post-fusion state (*Figure 1*), especially reorientation of D3 to fold over D1 and D2 in the trimer, which is required for fusion (*Figure 1*). Direct blocking of fusion has been demonstrated with antibodies to D3 of viral type II fusion proteins that are structurally homologous to HAP2 (*Austin et al., 2012*; *Zhao et al., 2016*; *Li et al., 2018*).

To test this idea, we superimposed D3 from *P. berghei* HAP2 on D3 from the trimeric fusion conformation of HAP2 in other species (*Figure 5B*). The *C. reinhardtii* and *A. thaliana* D3 structures superimpose well, despite only 8% and 17% identity with *P. berghei* HAP2 D3, respectively. This identity is too low for immunological crossreactivity. Superposition showed that the F-G loop recognized by mAb 2/6.14 is buried in D3 interfaces in the postfusion state trimer. The mAb 2/1.12 epitope also substantially overlaps with the D3 postfusion state trimer interfaces in the C strand. Moreover, both antibody epitopes overlap in the D strand with sites buried in the postfusion conformation. The incompatibility of Fab binding and conformational change to the postfusion state is shown by burial of the Fabs in superpositions on one monomer in a postfusion trimer of *C. reinhardtii* HAP2 (*Figure 4—figure supplement 2*). Thus, binding of either of these mAbs to D3 would block folding back of D3 in the fusion state (*Figure 1C*).

Unfortunately, we can only speculate on why mAb 2/6.14 and not mAb 2/1.12 blocked conversion of macrogametes to ookinetes. mAb 2/6.14 is differentiated from the other 4 mAbs studied here by its ability to completely react with the monomeric HAP2 ectodomain. Although all five mAbs reacted with D3 with nanomolar EC50 values by ELISA and pulled down similar amounts of isolated D3 by immunoprecipitation, only mAb 2/6.14 fully immunoprecipitated the ectodomain. SPR measurements further showed that the affinity of mAb 2/1.12 was substantially lower for the ectodomain than for D3, but only when the ectodomain was used as analyte and not when immobilized on the chip, suggesting that only ~10% of the ectodomain was active in binding to mAb 2/1.12. Gel filtration showed that the ectodomain complexed well with mAb 2/6.14, whereas only a minor fraction of the ectodomain complexed with mAb 2/1.12. Thus, three independent methods show that only a minor fraction of the HAP2 ectodomain reacts with mAb 2/1.12, and one of these methods, immunoprecipitation, showed that the other three mAbs behaved like mAb 2/1.12. While all 5 mAbs gave clear-cut staining of *P. berghei* microgametes, the microgametes had been fixed in 4% paraformaldehyde and incubated overnight in this solution, which denatures many proteins, prior to staining. Among many possible explanations for this discrepancy, perhaps mAb 2/1.12 recognizes an epitope that is available in D3 but is masked or destabilized by association with D1 and D2 in the HAP2 ectodomain, and this inhibition by D1 and D2 is relieved by partial denaturation during fixation by paraformaldehyde.

The scope of this study was to investigate the effectiveness of HAP2 D3 as an immunogen for transmission-blocking malaria vaccines. Nonetheless, our study also provides some useful guideposts for future use of the HAP2 ectodomain in transmission-blocking vaccines. Most eukaryotic extracellular proteins are highly dependent on N-glycosylation for stability and expression; however, lack of N-glycosylation sites in D3 appeared to have little or no effect on expression yield, which favors the hypothesis that N-glycans are not added to HAP2. Unfortunately, efficiently expressing eukaryotic extracellular proteins with complex multi-domain structures, multiple disulfide bonds, and no N-glycosylation requires refolding from *E. coli*, which is highly challenging. Therefore, the best approach for successfully expressing the HAP2 ectodomain may be to use eukaryotic hosts and to mutationally remove N-glycosylation sequons, as was done here for D3 but not for the ectodomain.

In conclusion, we find that D3 of HAP2 can elicit antibodies that block transmission of malaria. Furthermore, we have obtained a mAb that blocks transmission, determined the first structure of a fragment of HAP2 in *Plasmodium*, and determined the structure of HAP2 of D3 in *Plasmodium* in complex with antibodies that either block or do not block transmission. We have confirmed the principle that HAP2 D3 can elicit transmission-blocking antibodies. On the other hand, we have also identified limitations of D3 as an immunogen because some antibodies to D3 did not react well with the HAP2 ectodomain. We also outline a possible strategy for obtaining improved expression and more native folding of the HAP2 ectodomain as an alternative immunogen for transmission-blocking immunity. This proposed approach is validated by our EM studies showing we can isolate a monomeric, pre-fusion state of the HAP2 ectodomain. Furthermore, stabilizing the pre-fusion state of HAP2, as successfully done for the respiratory syncytial virus fusion protein (*McLellan et al., 2013*)

and SARS-CoV-2 spike protein (*Hsieh et al., 2020*), may not only increase expression but also efficacy in inducing neutralizing antibodies.

# Materials and methods

## Key resources table

| Reagent type (species) or resource | Designation | Source or reference | Identifiers | Additional information |
|---|---|---|---|---|
| Genetic reagent (*Drosophila*) | EXpreS2 transfection reagent | ExpreS2ion Biotechnologies | Catalog NO: 95-055-075 | https://expression systems.com/product/expres2-tr-transfection-reagent/ |
| Cell line (*Drosophila*) | *Drosophila melanogaster* Schneider S2 | ExpreS2 cells | ExpreS2ion Biotechnologies | |
| Transfected construct (*Drosophila*) | ET15S2 vector | *Feng et al., 2018* | ExpreS2ion Biotechnologies | Modified from the pExpreS2-2 vector, Includes N-terminal secretion signal from Hspa5 and C-terminal His8 tag |
| Cell line (*Homo-sapiens*) | Expi293F cells | Thermo Fisher Scientific | Catalog NO: A14527 | |
| Transfected construct (Expi293F cells) | pD2529-CAG vector | This paper | Atum, Newark, CA | |
| Strain, strain background (*M. musculus*) | CB6F1 | The Jackson Laboratory | | CB6F1 mice were immunized with the glycan-shaved D3 protein |
| Strain, strain background (*P. berghei*) | *P. berghei* | This paper | *P. berghei* – WT strain ANKA 2.34 | Strain maintained in Dr. Andrew Blagborough lab |
| Strain, strain background (Mosquitoes) | Female *Anopheles stephensi* | This paper | SDA 500 strain | Strain maintained in Dr. Andrew Blagborough lab |
| Gene (*P. berghei*) | *P. berghei* HAP2 ectodomain | UniProt | Q4YCF6.1 | All constructs were codon-optimized for mammalian cells |
| Gene (*P. falciparum*) | HAP2 D3 (aa:479–626) | NCBI | XP_001347424.1 | All constructs were codon-optimized for mammalian cells |
| Gene (*P. knowlesi*) | HAP2 D3 (aa:492–607) | NCBI | XP_002258781.1 | All constructs were codon-optimized for mammalian cells |
| Gene (P.vivax) | HAP2 D3 (aa:492–608) | GenBank | SGX77070.1 | All constructs were codon-optimized for mammalian cells |
| Gene (*P. malariae*) | HAP2 D3 (aa:498–613) | GenBank | SCN12177.1 | All constructs were codon-optimized for mammalian cells |
| Gene (*P. ovale*) | HAP2 D3 (aa:493–608) | GenBank | SBS88209.1 | All constructs were codon-optimized for mammalian cells |
| Antibody | Anti-His (Rabbit polyclonal) | Cell Signaling | Catalog NO:2,365 S | (0.4 µg/ml) |
| Antibody | Rabbit anti-alpha tubulin (Rabbit polyclonal) | Abcam | Catalog NO:AB18251 | (1:1000) |
| Antibody | Goat anti-Mouse IgG (H + L) Cross-Adsorbed Secondary Antibody, Alexa Fluor 488 (Goat polyclonal) | Molecular Probes/ThermoFisher | Catalog NO: A-11001 | (25 µg/ml) |
| Antibody | Goat anti-Rabbit IgG (H + L) Cross-Adsorbed Secondary Antibody, Alexa Fluor 594 (Goat polyclonal) | Molecular Probes/ThermoFisher | Catalog NO:A-11012 | (25 µg/ml) |
| Software, algorithm | XDS | https://strucbio.biologie.uni-konstanz.de/xdswiki/index.php/Xds | | Diffraction data was processed with XDS |

*Continued on next page*

*Continued*

| Reagent type (species) or resource | Designation | Source or reference | Identifiers | Additional information |
| --- | --- | --- | --- | --- |
| Software, algorithm | Phenix | https://www.phenix-online.org/ | | The structure was solved by molecular replacement with Phaser in the Phenix suite |
| Software, algorithm | CCP4 | http://www.ccp4.ac.uk/ | | Refinement |

## Study design

This study was designed to provide insights for developing a vaccine to block or reduce malaria transmission. This objective was addressed first by generating monoclonal antibodies against the D3 fragment of *Plasmodium berghei* HAP2. CB6F1 mice were immunized with the glycan-shaved D3 protein. All in vitro characterization of binding properties was carried out after a detailed antibodies screening as described in Methods. We next tested the ability of the D3 mAbs to inhibit *P. berghei* microgametes from fertilizing macrogametes and forming ookinetes in vitro. The results showed that all five antibodies reacted with microgametes, and that mAb 2/6.14 was outstanding for its ability to block P. berghei fertilization in vitro. Therefore, we decided to test mAb 2/6.14 for its ability to block transmission in vivo in mosquitoes. For in vivo characterization of the ability of mAb 2/6.14 *to* reduce malaria transmission, female *Anopheles stephensi* mosquitoes were randomized to group feed through membranes on blood from *P. berghei* infected mice mixed with mAb 2/6.14 or control IgG. After 14 days, the number of oocyst of infected mosquitoes was counted to evaluate the ability of mAb 2/6.14 to inhibit transmission.

## Ethical statement for animal studies

Mouse immunization was conducted in accordance with and was approved by Boston Children's Hospital Institutional Animal Care and Use Committee (IACUC) under protocol #14-06-2731. Animals were cared in compliance with the U.S. Department of Agriculture (USDA) Animal Welfare Act (AWA) and the Public Health Service (PHS) Policy on Humane Care and Use of Laboratory Animals.

Immunization and *Plasmodium berghei* infection procedures were performed in accordance with the UK Animals (Scientific Procedures) Act (PPL 70/8788) and were AWERB approved. The Office of Laboratory Animal Welfare Assurance covers all Public Health Service supported activities involving live vertebrates in the United States (no. A5634-01).

## HAP2 constructs, expression, and protein purification

*P. berghei* HAP2 amino acid sequence numbering is from RefSeq accession XP_022713330.1. D3 constructs utilized residues 477–621, 497–617, or 502–617 with a C-terminal His tag; only the 502–617 construct contained N516T, S533N and N539Q mutations to abolish N-glycosylation. HAP2 D3 construct sequence ranges and accessions in other species were *P. falciparum*, 479–626, XP_001347424.1; *P. knowlesi*, 492–607, XP_002258781.1; P. vivax, 492–608, SGX77070.1; *P. malariae*, 498–613, SCN12177.1; and *P. ovale,* 493–608, SBS88209.1. These D3 constructs were expressed either transiently in Expi293F cells using pD2529-CAG vector (Atum, Newark, CA), or in the case of the *P. berghei* HAP2 D3 477–621 construct and ectodomain 61–611 and 43–617 constructs, and the *P. falciparum* D3 479–626 construct in *Drosophila* Schneider S2 cells, were expressed exactly as described previously (*Feng et al., 2018*). All constructs were codon-optimized for mammalian cells. UniProt accession Q4YCF6.1, which was used for *P. berghei* HAP2 ectodomain construct codon optimization, was retrospectively discovered to be deleted for S206, which locates to the βd-strand of D2.2 in the alignment to *Chlamydomonas reinhardti* HAP2, at the opposite end of the ectodomain from D3. All other eight HAP2 accessions recovered from the nonredundant protein database with NCBI BLAST in 2020, including at least seven distinct *P. berghei* strains including ANKA, are identical to one another and differ from the Q4YCF6.1 sequence only at residue S206 and the signal sequence. This error has been reported to help@uniprot.org. Our recent experience is that, among hundreds of sequences of human and mouse extracellular proteins of similar length, ~ 10% of UniProt but not RefSeq accessions have similar errors.

For purification, HAP2 fragments in 1 L of culture supernatant were adjusted to 1 mM NiCl$_2$ in D3 buffer (20 mM Tris, pH 8 and 300 mM NaCl) or ectodomain buffer (20 mM Tris, pH 8.5, 500 mM NaCl) and applied to 10 ml Ni-NTA-Agarose (Qiagen) columns. After washing with 15 mM imidazole in D3 or ectodomain buffer, the protein was eluted with 300 mM imidazole in the same buffers. Fragments were then subjected to gel filtration chromatography using Superdex 75 (GE life Sciences) in D3 or ectodomain buffer, concentrated, and frozen at –80 °C. Purification of *P. berghei* D3477–621 construct expressed stably in S2 cells was similar, except after the Ni-NTA step it was concentrated (1 mg/ml in 0.2 ml) and shaved with endoglycosidase D (10 µl, 100 unit) (New England BioLabs) for 16 hrs at 4 °C prior to gel filtration. Final yield was 22 mg/L culture supernatant. Yields for transiently expressed D3 constructs were ~5–8 mg/L culture supernatant. Ectodomain yields were ~0.2–0.3 mg/L culture supernatant.

## Monoclonal antibodies

CB6F1 mice (Charles River, Wilmington, MA) were immunized intraperitoneally with 20 µg of N-glycan shaved *P. berghei* HAP2 D3 (residues 477–621) in PBS and complete Freund's adjuvant (Sigma) and 3 weeks later with the same material in incomplete Freund's adjuvant (Sigma). Mice were boosted both intravenously and intraperitoneally 2 weeks later with 20 µg of the same antigen in PBS. Three days later, splenocytes were fused with the murine myeloma P3 × 63Ag8.653 (CRL 1580, ATCC, Rockville, MD) as described (*Springer, 1980*). Hybridomas supernatants were screened by ELISA in microtiter wells coated with the immunogen. Hybridomas that produced IgG mAbs were subcloned. Five produced antibodies specific for *P. berghei* HAP2 D3 and are characterized in Results. mAb 1/5.13 was found to be specific to the His-tag. It reacts with proteins with C-terminal His tags, but not with N-terminal His tags. It is fully active in ELISA, western blot and immunoprecipitation and has sensitivity comparable to the THE His Tag Antibody (GenScript) in ELISA and western blot (*Figure 2—figure supplement 2*).

For antibody purification, hybridoma cells were adapted to and expanded in serum free medium containing 1:2 vol/vol Cell MAB medium (Life Technologies) and HyClone CDM4MAB medium (GE Life Sciences). IgG was purified using protein G affinity chromatography (Invitrogen).

Heavy and light chain V region cDNA sequences of mAbs 2/6.14 and 2/1.12 were determined by Syd Labs (Natick, MA). The mAbs each have γ1 heavy and κ light chains and unique V regions as shown by BLAST searches. A somatic mutation in the 2/6.14 VL domain that introduced an N-glycosylation site into the framework region was reversed with an N74S mutation (mature protein numbering) in recombinantly expressed Fab and IgG. Fab or intact H chains used g-Blocks encoding the murine κ chain secretion signal peptide, VH and γ1 CH1 domain, with or without hinge, CH2 and CH3 domains, followed by a Gly-Ser linker and 6xHis tag, that were assembled in EcoRV-linearized pVRC8400 vector (*Barouch et al., 2005*) using NEBuilder HiFi DNA reagents and protocol (New England BioLabs). κ light chains used g-Blocks encoding the murine κ chain secretion signal peptide, VL domain, and CL domain that were similarly assembled with SapI-linearized pD2529-CAG vector (Atum, Newark, CA).

Fabs and IgGs were expressed in Expi293F cells co-transfected with H and L chain constructs in 2:1 ratios. Fab fragments were purified from culture supernatant by Ni-NTA affinity purification followed by Superdex 200 gel filtration chromatography. IgG was purified by protein G affinity chromatography. 2/6.14 IgG purified from Expi293F and hybridoma cell supernatants bound to *P. berghei* HAP2 D3 in ELISA with comparable EC50 values.

## Enzyme-linked immunosorbent assay (ELISA)

96-well ELISA plates (Costar) were coated overnight at 4 °C with 50 µl of purified, His-tagged HAP2 D3 at 5 µg/ml in 50 mM sodium carbonate buffer, pH 9.5 and blocked with 3% BSA for 90 min at 37 °C. 50 µl of diluted hybridoma supernatants or purified mAbs in triplicate were incubated for 2 hrs at 37 °C. After three washes, 50 µl of 1:10,000 diluted HRP-conjugated goat-anti-mouse IgG (H + L) (Abcam) was added. After 1 hr at room temperature and four washes, peroxidase substrate (Life Technologies) was added and after 10 min plates were read at 405 nM on an Emax plate reader (Molecular Devices). As a control, hybridoma medium or mAb dilution buffer (1% BSA in PBS) was substituted in the antibody binding step. For measuring binding of His-tagged D3 to immobilized mAbs, ELISA plates were coated with 5 µg/ml mAb, and HRP-anti-His (Penta-His Ab, Qiagen) was

used in the detection step. Titration curve fitting and EC50 measurements used the sigmoidal, 4 parameter logistic equation in GraphPad Prism 7.

## Surface plasmon resonance (SPR)

Purified HAP D3 (residues 502–617 with N516T, S533N and N539Q mutations) or ectodomain (residues 43–617) fragments or Fabs 2/1.12 or 2/6.14 were either used as analytes or amine immobilized on a CM5 chip in a Biacore 3000 (GE Healthcare) according to the manufacturer's instructions. For immobilization, protein was diluted to 5 µg/ml in 0.15 M NaCl, 20 mM Hepes pH 7.4, and injected at 10 µL/min. The surface was regenerated with a 10- to 60 s pulse of 18 mM HCl at the end of each cycle to restore resonance units to baseline. Kinetics and affinity analysis were performed with SPR evaluation software version 4.0.1 (GE Healthcare). A 1:1 Langmuir binding model, with or without a conformational change model, was applied for experimental data fitting, and kinetic parameters were fit globally to sensorgrams at different analyte concentrations.

## Bio-layer interferometry (BLI)

Bio-layer interferometry (*Wallner et al., 2013*) experiments were performed on a ForteBio Octet RED384 instrument using anti-mouse Fc capture (AMC, 18–5090) sensors. The reaction was measured on a 384-well plate (working volume of 25 µL) in 20 mM Tris-HCl, pH7.5, 150 mM NaCl, 0.01%BSA, 0.02% Tween 20 (Assay buffer). Biosensors were hydrated in assay buffer for 10 min before starting the measurements. Each biosensor was sequentially moved through five wells with different components: (1) Assay buffer for 1 min in baseline equilibration step; (2) 5 µg/ml (33.3 nM) 2/6.14 or 2/1.12 IgG for 3 min for immobilization of the antibodies onto the biosensor; (3) Assay buffer for 3 min for another baseline equilibration; (4) indicated concentrations of PbHAP2 ectodomain (aa:43–617) or PbD3(aa: 477–621) for 10 min for the association phase measurement; and (5) Assay buffer for 10 min for the dissociation phase measurement. Each biosensor has a corresponding assay buffer reference sensor that went through the same five steps. Kinetics and affinity analysis were performed with Octet RED384 Data Analysis 11.0. A 1:1 Langmuir binding model was applied for experimental data fitting, and kinetic parameters were fit globally to different analyte concentrations for each IgG and HAP2 combination, with $k_{on}$ and $k_{off}$ as shared fitting parameters and maximum response ($R_{max}$) as individual fitting parameter.

## Immunoprecipitation and western blot

Culture supernatants or purified HAP2 D3 and ectodomain fragments diluted in TBS (25 mM Tris, pH 8, 300 mM NaCl) containing 0.5% BSA were incubated with mAbs or control non-binding IgG overnight at 4 °C. Immunocomplexes were pulled down with protein G beads by incubation at 4 °C for 2 hr with rotation. Beads were washed 3 times with 1 ml TBS and bound proteins eluted in 1 x Laemmli sample buffer containing 5% β-mercaptoethanol and subjected to reducing SDS-polyacrylamide gel electrophoresis. Blots were probed with polyclonal rabbit anti-His (0.4 µg/ml, Cell Signaling), followed by incubation with HRP-conjugated goat-anti-rabbit (GE Healthcare) and chemiluminescence imaging using LAS-4000 system (Fuji Film). Quantitation of protein bands used ImageJ software.

## Assays with *P. berghei* Gametes

For immunofluorescent staining of microgametes, tail blood (10 µl) from a mouse with high gametocytaemia was mixed with 10 µl of ookinete medium (RPMI1640 containing 25 mM HEPES, 20% FCS, 100 µM xanthurenic acid pH 7.4) to stimulate emergence of male gametes (exflagellation), which was monitored microscopically as described (*Sebastian et al., 2012*). Exflagellating gametocytes were fixed in solution for 15 min by diluting formalin to 4% paraformaldehyde, then added to poly-L-lysine-coated slides and incubated at 4 °C overnight. For staining, mAbs or control mouse IgG UPC10 (M7769, Sigma Aldrich) (2 µg in 4 µl of 3% BSA/PBS) were added to washed slides; rabbit anti-alpha tubulin (Abcam AB18251) was at 1:1000 dilution. Following washes, Alexa Fluor-488 anti-mouse IgG and Alexa Fluor-594 anti-rabbit IgG secondary antibodies (Molecular Probes) were added at 25 µg/ml before mounting in VectaShield with DAPI (Vector Laboratories). Fluorescence images were obtained using an epifluorescence 100 x objective on a Nikon Eclipse Ti microscope. Image handling used Adobe Photoshop CC.

In vitro ookinete conversion was assayed as described (*Angrisano et al., 2017*; *Blagborough et al., 2013*). Briefly, infected mouse blood containing *P. berghei* (strain ANKA 2.34) female and male gametocytes (20 μl) was mixed with ookinete medium (100 μl) containing HAP2 D3 antibodies or control IgG (500 and 250 μg/ml final concentration) and incubated at 19 °C. After 24 hr, cultures were incubated with Cy3-conjugated Pbs28 mAb 13.1, which stains both ookinetes and unfertilized macrogametes (*Winger et al., 1988*), for 20 min on ice. Larger, elongated, banana-shaped ookinetes were distinguished from smaller, round macrogametes and counted by fluorescence microscopy. Conversion rates were calculated as % ookinetes /(macrogametes + ookinetes). Inhibition of ookinete conversion was expressed as the percentage reduction in ookinete conversion with respect to the negative control IgG at the same concentration.

In vivo transmission-blocking activity of HAP2 antibodies was assayed using standard membrane feeding assay (SMFA) as described (*Blagborough et al., 2013*). Briefly, heparinized *P. berghei* infected blood containing gametocytes was mixed with HAP2 antibodies or negative control IgG. Female *Anopheles stephensi* (SDA 500 strain) were starved for 24 hr and then fed on the mixtures using membrane feeders ( > 50 mosquitoes per each blood-antibody feed). Twenty-four hr later, unfed mosquitoes were removed. Mosquitoes were maintained on 8% (w/v) fructose, 0.05% (w/v) p-aminobenzoic acid at 19–22°C and 50–80% relative humidity. Day 14 post-feeding, mosquito midguts were dissected, and oocyst numbers per midgut in each mosquito was determined by phase contrast microscopy. Reductions in oocyst intensity (number of oocysts/midgut) and prevalence (number of infected over total mosquitoes fed) in the presence of an HAP2 antibody were calculated with respect to the negative control IgG present at the same concentration in the feeds.

## Crystallization and structure determination

The *P. berghei* D3 (502–617) construct with N516T, S533N and N539Q mutations to abolish N-glycosylation was used for crystallization. D3 and Fab were mixed in 1:1.3 molar ratios and complexes were isolated by gel filtration. Complexes were crystallized at 20 °C by hanging-drop vapor diffusion with equal volumes of complex and well solution. The Fab 2/1.12 complex (4.5 mg/ml) was crystallized with 0.2 M ammonium sulfate, 25% PEG 3350, 0.1 M Bis-Tris pH5.5. The 2/6.14 Fab complex (7.5 mg/ml) was crystallized with 0.3 M proline, 22% PEG 1500, 0.1 M HEPES pH7.5; crystals were dehydrated by soaking in solutions that had the starting concentrations of components in the protein and reservoir solutions while raising the concentration of PEG 1500% to 31% in 3% steps. Crystals of Fab 2/1.12-D3 and Fab 2/6.14-D3 were cryo-protected with reservoir solution containing 15% glycerol or 15% ethylene glycol, respectively, in 2 steps of 5% and 10% increase and plunged in liquid nitrogen. Data were collected at 100 K on GM/CA beamline 23-IDB at the Advanced Photon Source (Argonne National Laboratory) and processed with XDS (*Kabsch, 2001*). Structures were refined with PHENIX, built with Coot (*Emsley and Cowtan, 2004*) and validated with MolProbity (*Davis et al., 2007*). Figures were made with PyMol.

The 2/1.12 Fab-D3 complex structure was solved by molecular replacement in the Phenix suite (*Adams et al., 2010*) using a Fab search model (PDB ID 2A6D). The 2/6.14 Fab-D3 complex structure was solved by molecular replacement using D3 from the 2/1.12 Fab complex, the H chain of PDB ID 1IGC, and the L chain of PDB ID 5AZ2. Each crystal structure has two complexes in the asymmetric unit.

During model building and refinement of the 2/6.14 Fab complex, the Fab constant domains of one complex had good density, but the variable domains and D3 had broad but continuous density that was difficult to trace. In contrast, all domains of the other complex were easily traced. Furthermore, refinement remained stuck. Alternative space groups including those with lower symmetry or use of twin rules provided no improvement. We then realized that in the troublesome D3-Fab complex in the asymmetric unit, two alternative conformations were present for D3, VH, and VL, whereas CH1 and CL had a single conformation. The transition between dual and single conformations occurred at the elbows between VH and CH1 and between VL and CL; that is the two conformations differed in elbow angle. In further refinement, the data cutoff was changed from 2.4 to 2.8 Å, and we largely treated each of the dual conformations of D3, VH1, and VL as rigid bodies based on their structure in the single conformation of the other D3-Fab complex.

The alternate conformations allow the VL domain in chain B and the D3 domain in chain C to pack against alternate conformation symmetry mates in different complexes along adjacent edges of the

unit cell. Symmetry mates with the B conformation severely clash, while those with the distinct conformations A and B have good lattice contacts. In contrast, symmetry mates with conformation A are too far apart to provide stabilizing lattice contacts. Thus, clashes in one conformation together with a lack of stabilizing lattice contacts in the other conformation may have driven the formation of a crystal lattice with dual conformations of VL, VH, and D3 in one of two Fab-D3 complexes in the asymmetric unit.

## Negative stain electron microscopy

Each *P. berghei* HAP2 ectodomain construct (residues 43–617 or 61–611), with or without Fab at a molar ratio of 1:1.3, was subjected to Superdex 200 gel filtration in 20 mM Tris-HCl, pH8, 500 mM NaCl. Peak fractions were adsorbed to glow-discharged carbon-coated copper grids, washed with deionized water, and stained with freshly prepared 0.75% uranyl formate. Images were acquired with an FEI Tecnai-T12 transmission electron microscope at 120 kV and a nominal magnification of 52,000×. About two thousand particles were picked interactively and subjected to 2D alignment, classification and averaging using RELION-2.1 (*Zivanov et al., 2018*). Selected EM class averages were masked and cross-correlated using EMAN2 (*Ludtke et al., 1999*) with 2D projections generated at 2° intervals from the 2/6.14 Fab-D3 complex crystal structure filtered to 25 Å.

## Cell lines

Expi293F cells and *Drosophila* Schneider S2 cells were used to express proteins. Expi293F cells were purchased from Thermo Fisher Scientific. The S2 cells were obtained from ExpreS2ion Biotechnologies. All cells tested negative for mycoplasma.

## Acknowledgements

We thank Margaret Nielsen for illustrations. We thank Kelly L Arnett from Center for Macromolecular Interactions of Harvard Medical school for training and consultation on Bio-layer interferometry measurement. This work was supported by NIH grant 5R01AI95686 (TAS and CL) the Kidder Fund from Boston Children's Hospital (TAS), Medical Research Council grant MR/N00227X/1, Isaac Newton Trust, Alborada Fund, Wellcome Trust ISSF, University of Cambridge JRG Scheme, GHIT, Rosetrees Trust and the Royal Society (AMB).

## Additional information

### Funding

| Funder | Grant reference number | Author |
| --- | --- | --- |
| National Institutes of Health | R01AI95686 | Chafen Lu<br>Timothy A Springer |
| Boston Children's Hospital | Kidder Fund | Timothy A Springer |
| Medical Research Council | MR/N00227X/1 | Andrew M Blagborough |
| Isaac Newton Trust | | Andrew M Blagborough |
| Alborada Trust | | Andrew M Blagborough |
| Wellcome Trust | | Andrew M Blagborough |
| University of Cambridge | | Andrew M Blagborough |
| Global Health Innovative Technology Fund | | Andrew M Blagborough |
| Rosetrees Trust | | Andrew M Blagborough |
| Royal Society | | Andrew M Blagborough |

The funders had no role in study design, data collection and interpretation, or the decision to submit the work for publication.

## Author contributions
Juan Feng, Conceptualization, Formal analysis, Investigation, Methodology, Validation, Writing - original draft; Xianchi Dong, Formal analysis, Methodology; Adam DeCosta, Investigation; Yang Su, Data curation; Fiona Angrisano, Andrew M Blagborough, Writing – review and editing, Investigation; Katarzyna A Sala, Investigation, Methodology; Chafen Lu, Investigation, Supervision, Writing – review and editing, Conceptualization; Timothy A Springer, Conceptualization, Funding acquisition, Project administration, Supervision, Writing – review and editing

## Author ORCIDs
Andrew M Blagborough http://orcid.org/0000-0002-5257-8475
Chafen Lu http://orcid.org/0000-0002-3954-4836
Timothy A Springer http://orcid.org/0000-0001-6627-2904

## Decision letter and Author response
Decision letter https://doi.org/10.7554/eLife.74707.sa1
Author response https://doi.org/10.7554/eLife.74707.sa2

# Additional files

## Supplementary files
• Transparent reporting form

## Data availability
Protein database accession IDs are 7LR3 for 2/6.14-Pb HAP2 D3 complex and 7LR4 for 2/1.12-Pb HAP2 D3 complex. Correspondence and requests for materials should be addressed to CL and TAS.

The following dataset was generated:

| Author(s) | Year | Dataset title | Dataset URL | Database and Identifier |
| --- | --- | --- | --- | --- |
| Feng J, Dong XC, Springer TA, Lu CF | 2021 | Complex of Fab 2/6.14 with domain 3 of P. berghei HAP2 | https://www.rcsb.org/structure/7LR3 | RCSB Protein Data Bank, 7LR3 |
| Feng J, Dong XC, Springer TA, Lu CF | 2021 | Complex of Fab 2/1.12 with domain 3 of P. berghei HAP2 | https://www.rcsb.org/structure/7LR4 | RCSB Protein Data Bank, 7LR4 |

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
