## [Editor Report]

This study describes the generation of monoclonal antibodies targeting the fusogen HAP2 from the malaria parasite that is required for parasite transmission to mosquitoes. Using structural approaches in combination with biophysical methods, the authors provide insights into the differences in transmission blocking potencies for different monoclonal antibodies, which may inform the design of HAP2 as a potential vaccine candidate.

---

## [Decision Letter]

**Decision letter after peer review:**

Thank you for submitting your article "Structural basis of malaria transmission blockade by a monoclonal antibody to gamete fusogen HAP2" for consideration by *eLife*. Your article has been reviewed by 3 peer reviewers, one of whom is a member of our Board of Reviewing Editors, and the evaluation has been overseen by Dominique Soldati-Favre as the Senior Editor. The following individual involved in review of your submission has agreed to reveal their identity: Wai-Hong Tham (Reviewer #3).

Essential revisions:

All three reviewers agree that this is an interesting and important study, and that the work is very well conducted and presented. They also identified a number of points, as detailed below in the recommendations to the authors, which could be addressed in the text to strengthen and/or clarify the manuscript, without the need for additional experiments. Specifically, the reasons for discrepancy between binding to the recombinant ectodomain versus the native protein on gametes for the non-neutralizing mAbs could be discussed in more details. It could be due either to how the parasites are treated or due to the state of the ectodomain, as suggested by the observation that a fraction of the protein binds with good affinity, while a larger fraction doesn't.

*Reviewer #1:*

This is an interesting study describing the generation of monoclonal antibodies against the HAP2 protein from P. berghei, and providing functional and structural insights in antibody-mediated inhibition of parasite transmission. HAP2 is an evolutionary conserved gamete fusogen and a known transmission-blocking vaccine target. Here the authors produced recombinant forms of the D3 domain of P. berghei HAP2 protein and generated five monoclonal antibodies that recognize the recombinant protein as well as the native protein in P. berghei gametes. One of the antibody inhibits HAP2 function, as shown by reduced ookinete production in vitro and reduced parasite transmission to mosquitoes. The antibodies show some degree of cross-reactivity towards HAP2 from other Plasmodium species, except *P. falciparum*. The authors resolved the structure of the D3 domain bound to two of the antibodies, including the one with neutralization potential. The structure of the PbHAP2 D3 domain appears similar to known HAP2 structures from other organisms. The data reveal that the two antibodies recognize distinct epitopes on the protein. Strikingly, only the neutralizing mAb can recognize the full-length HAP2 domain.

Strengths

– This work provides novel antibodies against HAP2 from the malaria parasite, and the first structural data for a Plasmodium HAP2 protein.

– Some degree of cross-reactivity was observed with HAP2 from other Plasmodium species, providing potentially interesting tools to study HAP2 function.

– The antibodies vary in their neutralization potential, which may be useful to investigate the mechanisms of transmission blocking.

Limitations

– The work was performed with a rodent species (P. berghei). Although the antibodies cross-react, at least in part, with HAP2 from other species, they do not recognize *P. falciparum* protein, which limits direct translation of the conclusions to the deadliest human malaria parasite.

– The only inhibitory mAb is the one that recognizes the full-length protein, so it is not clear to which extent the D3 domain represents an attractive target as compared to other domains of the protein. It is not clear how the data will inform vaccine design targeting HAP2.

– The fact that all five antibodies recognize the native protein in gametes yet only one mAb efficiently recognizes the full-length recombinant ectodomain is puzzling.

– The authors propose a model of action of the blocking antibody, but the mechanisms of neutralization are not addressed in the current manuscript.

I am not an expert in structural biology so I cannot evaluate the quality of the structural data.

– The authors propose a model of action of the blocking antibody in Figure 1, yet the mechanisms of neutralization are not addressed in the current manuscript. In particular, what is the evidence that the 2/6.14 mAb can exert Fc-dependent immune effector functions in the mosquito? Also, at which step does this antibody block gamete fusion? According to the model in Figure 1, one would expect that binding of male and female gametes would not be affected by the antibody. This could be tested experimentally.

– It is not clear how the data will inform vaccine design targeting HAP2. Only the 2/6.14 mAb shows neutralization potential, and strikingly it is also the only one that efficiently recognizes the full-length protein, so it is not clear to which extent the D3 domain represents an attractive target as compared to other domains of the protein.

– The fact that all five antibodies recognize the native protein in gametes yet only one mAb efficiently recognizes the full-length recombinant ectodomain is puzzling. The authors discuss that prolonged fixation is responsible for alteration of the protein in the IFA. They should test shorter fixation conditions, or test binding of the antibodies without fixation, to analyze whether the mAbs bind to the gamete surface, as would be expected at least for mAb 2/6.14 based on the functional data.

– Could structural modelling provide possible explanations for the differential cross-reactivity of the PbHAP2 mAbs against HAP2 from other species, especially *P. falciparum*?

– The data show that the mAb 2/1.12 does not inhibit ookinete conversion in vitro (Figure 3B) but can it block transmission to mosquitoes?

*Reviewer #2:*

HAP2 is a protein required by the sexual stages of the malaria parasite to fuse and to complete fertilisation. It is therefore thought to be a transmission-blocking vaccine candidate. If a mosquito takes up antibodies which target HAP2 with the blood-meal, then they can prevent parasite gamete fusion in the mosquito midgut and can block transmission. This manuscript explores whether the D3 domain of HAP2 may be a valuable transmission blocking immunogen.

Based on analogy to viral fusogens, the authors decided to focus on domain D3 of HAP2, as antibodies against the equivalent domain of viral fusion proteins are effective against viral fusogens. Plasmodium berghei HAP2 D3 domain was used to raised monoclonal antibodies. These were structurally characterised and were tested for their ability to prevent transmission of malaria parasites in a P. berghei mouse malaria model.

To begin, the authors produced the D3 domain of 6 different Plasmodium species in insect cells, carefully ensuring correct expression and removal of glycans. They immunised mice wih the P. berghei protein and generated a panel of monoclonal antibodies. As the D3 domains from different species show 60-70% sequence identity, they assessed cross reactivity, showing two of the mAbs to pull down D3 domain from most Plasmodium species. While this is a nice result, it was disappointing that the worst cross-reactivity was with the D3 domain from the most-deadly *P. falciparum* species.

They next tested the effect of their mAbs. All five selected mAbs could label in vitro differentiated gametocytes. The mAbs were then tested in a transmission blocking assay, in which they were added to P. berghei parasites in mouse blood, which was fed to mosquitos. One mAb, 2/6.14 was found to block ookinete formation in the mosquitoes. While this mAbs showed efficacy, it is less effective than the 45.1 mAb which binds to the gametocyte surface protein Pfs48/45. While the authors 2/6.14 mAb is 44.5% effective at 50μg/ml, 45.1 shows 100% transmission blocking at 14μg/ml. Nevertheless, this is, to my knowledge, the best HAP2-targeting mAb to date.

The authors next determine structures of two antibodies – the most transmission blocking, 2/6.14 and the other cross-reactive antibody 2/1.12. These antibodies bind to different surface of the D3 domain. To understand why one is transmission blocking while the other is not, the authors then tested the binding of the panel of nAbs to full HAP2 ectodomain, containing all three domains. Here they find that only 2/6.14 interacts with the full ectodomain to an appreciable degree, perhaps explaining its ability to block transmission.

Finally, the authors model their D3:mAb structure onto a model of the post-fusion conformation of HAP2. They find that the mAb is incompatible with the adoption of this conformation. Could this block membrane fusion and account for the effect of the antibody in transmission blocking?

This study tests the hypothesis that the D3 domain of HAP2 acts can raise transmission-blocking antibodies and reveals the molecular determinants of binding of their best antibody. The manuscript is well conducted, careful and balanced. As to whether the D3 domain of HAP2 will make a great transmission blocking vaccine candidate, I would say that the jury is still out. The best antibody generated in this work is effective, but less so that those which target the transmission-blocking vaccine candidate Pfs48/45. It will be useful to see other studies in which D3 of *Plasmodium falciparum* HAP2 is used as a vaccine candidate and to see the efficacy of a larger panel of antibodies.

In the introduction, where the authors write "To mediate fusion, D3 of HAP2 folds over an inner trimeric core composed of D1 and D2 (Figure 1).", is this a proposal or model? If it has been proved, please cite the paper. If not, please indicate that this is a proposal.

In Figure 4A there is no structure shown for the two complexes – only opened up structures. This seemed strange and not intuitive. Could the structures of the complexes also be shown? Without this, it is very hard to understand how the two epitopes relate to each other from what is shown.

The authors also do not show the sulphate ions which they see in one of the structures. These ions appear to link the epitope and the antibody. How do they know that these are sulphate ions and are they likely to be physiologically relevant?

It would be good to move Figure S6 into the main manuscript as it helps explain develop a model for how the antibody works.

The number of clashes and RSRZ outliers are high for a 2.1Å structure for 7LR4. For 7LR3 the number of RSRZ outliers and Ramachandran outliers are both high. These structures appear to require more work in refinement.

*Reviewer #3:*

This was a great paper to read and straight-forward to review. All results are clearly stated without hyperbole and supported by their claims. We particularly appreciated that the authors also looked at the other HAP2 present in Plasmodium species for cross-reactive antibodies, examined the phenotype of the monoclonal antibodies using IFA, fertilization and transmission blocking, and used both X-ray crystallography and negative stain EM to provide structural insights into the mechanisms of inhibition (or lack of). It was also very useful to have experimental data showing that while one of the antibodies 2/1.12 could recognize D3 alone, it did not have the same level of recognition of the ectodomain. The Discussion covers this observation in an insightful way. The manuscript stands without further experiments.

---

## [Author Response]

Essential revisions:All three reviewers agree that this is an interesting and important study, and that the work is very well conducted and presented. They also identified a number of points, as detailed below in the recommendations to the authors, which could be addressed in the text to strengthen and/or clarify the manuscript, without the need for additional experiments. Specifically, the reasons for discrepancy between binding to the recombinant ectodomain versus the native protein on gametes for the non-neutralizing mAbs could be discussed in more details. It could be due either to how the parasites are treated or due to the state of the ectodomain, as suggested by the observation that a fraction of the protein binds with good affinity, while a larger fraction doesn't.

We have banged our collective heads against the wall trying to come up with a clear explanation (the PI's head is the most damaged) and cannot. All we can do is speculate. We do two things in Discussion. Instead of asking the question about why the discrepancy is present at the end of one paragraph, which may lead readers to expect an answer in the next paragraph, we delete that question and start the next paragraph by saying that we all we can do is speculate and offer one possibility at the end of that paragraph.

Reviewer #2:[…]In the introduction, where the authors write "To mediate fusion, D3 of HAP2 folds over an inner trimeric core composed of D1 and D2 (Figure 1).", is this a proposal or model? If it has been proved, please cite the paper. If not, please indicate that this is a proposal.

The whole conformational change was cited (a review) in the previous sentence.

In Figure 4A there is no structure shown for the two complexes – only opened up structures. This seemed strange and not intuitive. Could the structures of the complexes also be shown? Without this, it is very hard to understand how the two epitopes relate to each other from what is shown.

Thanks for pointing that out, we have added the two complex structures in the Figure 4A.

The authors also do not show the sulphate ions which they see in one of the structures. These ions appear to link the epitope and the antibody. How do they know that these are sulphate ions and are they likely to be physiologically relevant?

"The Fab 2/1.12 complex (4.5 mg/ml) was crystallized with 0.2 M ammonium sulfate, 25% PEG 3350, 0.1 M Bis-Tris pH5.5." These ions are highly unlikely to be physiologically relevant.

It would be good to move Figure S6 into the main manuscript as it helps explain develop a model for how the antibody works.

We do not know how the antibody works- It could work through blocking conformational change or it could simply opsonize the gametes for phagocytosis or it could agglutinate them.

The number of clashes and RSRZ outliers are high for a 2.1Å structure for 7LR4. For 7LR3 the number of RSRZ outliers and Ramachandran outliers are both high. These structures appear to require more work in refinement.

My groups prides itself in refinement. We are one of the few groups that reports MolProbity statistics in our crystal statistics tables. These show that our structures are in the 97-99th percentiles of all structures in the pdb of similar resolutions for clashes and geometry (which includes Ramachandran outliers). Thus the quality of our refinement is outstanding. Our style of refinement (and we usually push resolution more than others, because we do not throw out useful data, as recommended by Diedrichs and Karplus) tends to give high RSRZ outliers. We tend to build the mainchain through weak but continuous density and we do not truncate sidechains. This takes more care in refinement, provides a more useful model for biologists, but also gives higher RSRZ outliers. Additionally, the 7LR3 complex was difficult to refine as explained in Methods:

"During model building and refinement of the 2/6.14 Fab complex, the Fab constant domains of one complex had good density, but the variable domains and D3 had broad but continuous density that was difficult to trace. In contrast, all domains of the other complex were easily traced. Furthermore, refinement remained stuck. Alternative space groups including those with lower symmetry or use of twin rules provided no improvement. We then realized that in the troublesome D3-Fab complex in the asymmetric unit, two alternative conformations were present for D3, VH, and VL, whereas CH1 and CL had a single conformation. The transition between dual and single conformations occurred at the elbows between VH and CH1 and between VL and CL; i.e. the two conformations differed in elbow angle. In further refinement, the resolution cutoff was changed from 2.4 to 2.8 Å, and we largely treated each of the dual conformations of D3, VH1, and VL as rigid bodies based on their structure in the single conformation of the other D3-Fab complex."